# Measuring Diversity in Synthetic Datasets

**Yuchang Zhu** [† 1]   **Huizhe Zhang** [1]   **Bingzhe Wu** [2]   **Jintang Li** [3 4]   **Zibin Zheng** [3 4]   **Peilin Zhao** [5 6]   **Liang Chen** [* 1]
**Yatao Bian** [* 6 7]

## Abstract

Large language models (LLMs) are widely adopted to generate synthetic datasets for various natural language processing (NLP) tasks, such as text classification and summarization. However, accurately measuring the diversity of these synthetic datasets—an aspect crucial for robust model performance—remains a significant challenge. In this paper, we introduce `DCScore`, a novel method for measuring synthetic dataset diversity from a classification perspective. Specifically, `DCScore` formulates diversity evaluation as a sample classification task, leveraging mutual relationships among samples. We further provide theoretical verification of the diversity-related axioms satisfied by `DCScore`, highlighting its role as a principled diversity evaluation method. Experimental results on synthetic datasets reveal that `DCScore` enjoys a stronger correlation with multiple diversity pseudo-truths of evaluated datasets, underscoring its effectiveness. Moreover, both empirical and theoretical evidence demonstrate that `DCScore` substantially reduces computational costs compared to existing methods. Code is available at: https://github.com/bluewhalelab/dcscore.

## 1. Introduction

Large language models (LLMs) have shown exceptional performance across a range of fields, such as chatbots (Achiam et al., 2023), computer programming (Gu, 2023), and reasoning (Yuan et al., 2024). Inspired by their remarkable capacities, some research (Ye et al., 2022; Abdullin et al., 2024; Ding et al., 2024) employs LLMs as dataset generators to mitigate the shortage of training data. Although generated data facilitates model optimization, recent studies (Yu et al., 2024; Lee et al., 2023) suggest that a lack of diversity within the dataset—measured by the variation between samples (Long et al., 2024)—may lead to performance degradation in some scenarios. Although previous studies (Yu et al., 2024; Wang et al., 2022) leverage well-designed generation strategies to create highly diverse synthetic datasets, they consistently neglect to evaluate the diversity of these datasets. Additionally, a principled diversity evaluation metric serves not only to guide LLM generators in creating more diverse data but also extends its utility to data selection (Cao et al., 2023), quantifying augmentation performance (Yang et al., 2024a), and assessing mode collapse (Dan Friedman & Dieng, 2023). Thus, a diversity evaluation method for synthetic datasets is becoming increasingly important.

Since the diversity evaluation of synthetic datasets remains under-explored, a natural solution is to directly employ diversity evaluation methods from related fields, such as natural language processing (NLP) (Khurana et al., 2023) and machine learning (ML) (Jordan & Mitchell, 2015). Specifically, efforts to measure diversity within these domains can be summarized into three categories: *N-gram-based method* (Zhu et al., 2018; Mishra et al., 2020), *Reference-based method* (Heusel et al., 2017; Cífka et al., 2018), and *Transformation-based method* (Du & Black, 2019; Zhang et al., 2024). The n-gram-based method evaluates diversity through n-gram statistics, e.g., distinct-n (Li et al., 2015), focusing on textual form rather than semantic content. To align the diversity criteria with human judgment, the reference-based method has emerged as a promising alternative. This approach employs a reference distribution or data as an approximation of human judgment and calculates the similarity between the evaluated data and the reference data (Holtzman et al., 2019). However, collecting reference data can be both time-consuming and may introduce potential biases.

---

[†] Work done during an internship at Tencent AI Lab [*]Corresponding author  [1]School of Computer Science and Engineering, Sun Yat-sen University, Guangzhou, China [2]School of Artificial Intelligence, Shenzhen University, Shenzhen, China [3]School of Software Engineering, Sun Yat-sen University, Zhuhai, China [4]Zhuhai Key Laboratory of Trusted Large Language Models, Zhuhai, China [5]School of Artificial Intelligence, Shanghai Jiao Tong University, Shanghai, China [6]Tencent AI Lab, Shenzhen, China [7]Department of Computer Science, National University of Singapore, Singapore. Correspondence to: Yatao Bian <bianyt@comp.nus.edu.sg>, Liang Chen <chenliang6@mail.sysu.edu.cn>.

Drawing inspiration from deep representation learning (Butepage et al., 2017; Zhang et al., 2021), the transformation-based method evaluates diversity by first mapping the data into the representation space and then performing diversity summarization (Tevet & Berant, 2020). For data mapping, various embedding functions can be employed to facilitate the transformation, with a popular approach being the use of sentence transformers such as Sentence-Bert (Reimers, 2019) and SimCSE (Gao et al., 2021). Owing to the versatility of embedding functions, transformation-based methods can simultaneously consider extensive aspects, such as semantics and style, to encode data representations, thereby providing a more comprehensive evaluation of diversity. However, this type of method is hindered by suboptimal computational efficiency caused by the high-complexity diversity summarization, such as eigenvalue computation (Dan Friedman & Dieng, 2023).

In a nutshell, existing diversity evaluation methods in NLP and ML suffer from inherent limitations in the synthetic dataset evaluation scenario. To effectively evaluate the diversity of synthetic datasets, the following challenges must be tackled: (1) *Holistic Analysis*. The diversity evaluation of synthetic datasets is a holistic analysis task, necessitating consideration of the impact of each sample on the final evaluation results. (2) *Axiomatic Requirements*. Prior research has suggested several axioms that diversity metrics should ideally satisfy. To ensure reasonable evaluation, a well-established diversity evaluation should exhibit properties corresponding to these axioms. (3) *Lower Computational Costs*. As a growing volume of synthetic datasets, a diversity evaluation method with lower computational costs is highly desirable to ensure data quality and model performance.

To sum up, the essence of diversity is associated with the identification of differences between samples, and the ability to distinguish these differences is a key element in the classification process (Quine, 1969). Motivated by this observation, we propose a synthetic dataset diversity evaluation method from a classification perspective, namely, DCScore. Notably, DCScore tackles the three aforementioned challenges. Firstly, DCScore treats the evaluation of each sample in the synthetic dataset as an independent classification task, providing a holistic analysis. Secondly, theoretical verification demonstrates that DCScore satisfies four axioms outlined in (Leinster & Cobbold, 2012), including effective number, identical samples, symmetry, and monotonicity. Lastly, both empirical and theoretical evidence suggest that DCScore effectively evaluates the diversity of synthetic datasets, while demonstrating lower computational costs compared to existing methods. Our contributions can be summarized as follows:

- We propose DCScore, a classification-based diversity evaluation method for synthetic datasets. The core idea behind DCScore is to treat diversity evaluation as a sample classification task, enabling the capture of mutual relationships among samples.

- We theoretically validate that DCScore adheres to several intuitive axioms suggested by (Leinster & Cobbold, 2012), demonstrating its superiority. Additionally, we theoretically validate the lower complexity of DCScore under general kernels.

- Extensive experiments show that DCScore exhibits a stronger correlation with multiple diversity pseudo-truths compared to baseline metrics. We also perform a computational cost experiment to confirm the lower computational cost of DCScore.

## 2. Related Work

We give a brief literature review of diversity evaluation methods. Limited by space, further related works on LLM dataset generators and application of diversity evaluation methods can be found in Appendix A.

### 2.1. Diversity Evaluation Methods

With the development of LLMs as dataset generators, the diversity evaluation of synthetic datasets has become a challenging task and remains under-explored in recent evaluation studies (Liang et al., 2022; tatsu lab, 2023; Ribeiro et al., 2020). The most comparable diversity evaluation research can be traced back to studies in NLP and ML, which can be summarized into the n-gram-based method (Mishra et al., 2020), reference-based method (Heusel et al., 2017), and transformation-based method (Lai et al., 2020).

**N-gram-based Methods.** The n-gram-based method is the most popular lexical diversity evaluation method, leveraging n-grams to capture differences in sentence form (Yu et al., 2024). Commonly used n-gram-based diversity metrics include distinct n-grams (*distinct-n*) (Song et al., 2024), self-BLEU (Shu et al., 2019), and ROUGE-L (Wang et al., 2022; Padmakumar & He, 2023). However, this type of method only captures differences in text form, thereby overlooking differences in other aspects such as semantics and style.

**Reference-based Methods.** Diversity evaluation is a subjective task, leading to a reliance on human judgment. Consequently, the reference-based method evaluates diversity by comparing the distribution of the evaluated data to that of a reference dataset (Heusel et al., 2017). MAUVE (Pillutla et al., 2021) exemplifies this idea by employing a divergence-based metric to capture correlations with human judgment. Regarding the natural language inference (NLI) training set as the reference dataset, (Stasaski & Hearst, 2022) first trains an NLI model to infer the relationship between pairs of generated texts and then calculates diversity based on

these inference results. Due to the challenges in collecting reference datasets, recent studies (Kynkäänniemi et al., 2019; Le Bronnec et al., 2024) propose evaluating diversity through precision and recall. (Naeem et al., 2020) introduces density and coverage as solutions to the susceptibility of precision and recall to outliers. Despite these advancements, the reference-based method remains significantly constrained because of the need for reference datasets.

**Transformation-based Methods.** The transformation-based (Lee et al., 2023) method leverages well-designed models to generate representations of the evaluated data. Then, the diversity of these representations is summarized using techniques such as clustering (Du & Black, 2019) and eigenvalue computation (Dan Friedman & Dieng, 2023) of VendiScore (Dan Friedman & Dieng, 2023). In line with VendiScore (Dan Friedman & Dieng, 2023), RKE (Jalali et al., 2023) introduces an information-theoretic method for evaluating diversity in multimodal distributions, while FKEA (Ospanov et al., 2024) enhances the computational efficiency of both VendiScore and RKE. Owing to the superior performance of representation learning, this type of method considers various aspects of the evaluated data, including semantics, form, and style, offering greater flexibility compared to the other two methods. However, its dependence on high-complexity summarization techniques, such as eigenvalue computation, limits its scalability in evaluating the diversity of synthetic datasets.

In summary, existing methods primarily focus on NLP and ML fields and are challenging to apply directly to synthetic dataset diversity evaluation. Different from the above-mentioned studies, our work is dedicated to the holistic diversity evaluation of synthetic datasets. Additionally, to ensure flexible evaluation, our work aims to evaluate diversity-sensitive components that impact the performance of trained models in terms of diversity.

## 3. Preliminaries

### 3.1. LLM as a Dataset Generator

Since the exceptional performance of LLMs, previous works (Dai et al., 2023; Yoo et al., 2021) employ LLMs as a dataset generator or for data augmentation purposes. LLMs significantly reduce the cost of label annotation and data collection (Tan et al., 2024), and in several tasks, even outperform human annotators (Gilardi et al., 2023). While some studies attempt to use LLMs to generate datasets from scratch, it is a challenging task for LLMs. In most cases, a pre-trained LLM, denoted as $\mathcal{M}$, takes the data $\mathcal{D}_{sup}$ to be augmented and the generation task $T$ as input, and outputs the augmented dataset $\mathcal{D}$. Formally, this process can be formulated as follows:

$$\mathcal{D} \leftarrow \mathcal{M}(T, \mathcal{D}_{sup}) \tag{1}$$

*Table 1.* Categories of $\mathcal{D}_{sup}$. In the column of "Examples", texts belonging to $\mathcal{D}_{sup}$ are highlighted in gray , while texts associated with $T$ are marked using an underline.

| Generations | $\mathcal{D}_{sup}$ | Examples |
|---|---|---|
| $\{x_i\} \rightarrow \{y_i\}$ | $\{x_i\}$ | **Question:** It's a boring movie. The sentiment of the movie review is 
 **Answer:** Negative. |
| $\{y_i\} \rightarrow \{x_i\}$ | $\{y_i\}$ | **Question:** The movie review in positive sentiment is 
 **Answer:** Good film! |
| $\{\mathcal{T}_{seed}\} \rightarrow \{\mathcal{T}_i\}$ | $\{\mathcal{T}_{seed}\}$ | **Question:** Following are examples of movie review and their sentiment labels. Generate samples according to these examples. 
 Example 1: A boring movie. (Negative); 
 Example 2: Oh, wonderful movie! (Positive). 
 **Answer:** A meaningful movie. (Positive) |

where $T$ can be texts describing the generation task, such as annotation. $\mathcal{D}_{sup}$, which comprises a small number of seed samples or unlabeled inputs, serves as supplementary materials to facilitate data augmentation. For example, we want LLMs to perform an annotation task for sentiment labeling, such as determining whether the sentiment is positive or negative. If we assume $\mathcal{D}_{sup}$ to be "*It's a boring movie.*", the description of $T$ could be "*The sentiment of the movie review is*". $\mathcal{D} = \{\mathcal{T}_i\}_{i=1}^n = \{(x_i, y_i)\}_{i=1}^n$ is the generated dataset with $n$ samples, where $\mathcal{T}_i = (x_i, y_i), x_i,$ and $y_i$ are the input-output sample, the input text, and the output text, respectively. Let $T_{\mathcal{D}}$ denote the downstream task of $\mathcal{D}$, when $T_{\mathcal{D}}$ is the question-answering task, $x_i$ and $y_i$ represent the question and answer, respectively.

It is worth noting that not all components in $\mathcal{D}$ are generated by $\mathcal{M}$, which is related to the category of $\mathcal{D}_{sup}$. As shown in Table 1, $\mathcal{D}_{sup}$ can be divided into three categories, namely input text, output text, and seed samples. In Table 1, "$\rightarrow$" and $\mathcal{T}_{seed}$ represent the direction of generation and seed samples, respectively. For example, "$x_i \rightarrow y_i$" signifies that, given input text $x_i$ denoted as $\mathcal{D}_{sup}$, $\mathcal{M}$ processes $\mathcal{D}_{sup}$ and $T$, generating the output text $y_i$.

### 3.2. Problem Formulation

The diversity evaluation of the dataset is a sample richness evaluation problem. Based on the generation scenarios presented in Table 1, we find that in certain downstream tasks, the diversity of some components in the synthetic dataset does not influence the performance of the trained models. Conversely, the diversity of other components significantly impacts model performance. We refer to these components, whose diversity influences performance, as *diversity-sensitive components*, denoted as $\mathcal{T}_i$. For example, the input text $x_i$ in sentiment classification tasks is the diversity-sensitive component. Conversely, the output text $y_i$, which represents the sentiment label of the sample and is typically a numerical label (e.g., 0 or 1), does not influence model performance in terms of diversity. Thus, the output text cannot be considered as the diversity-sensitive

component. It should be underscored that diversity-sensitive components vary across downstream tasks. The diversity evaluation of synthetic datasets can be transformed into the diversity evaluation of diversity-sensitive components.

Given a synthetic dataset $\mathcal{D} = \{\mathcal{T}_i\}_{i=1}^n$, we define $\{\tilde{\mathcal{T}}_i\}_{i=1}^n$ as a collection of diversity-sensitive components. The problem of diversity evaluation of $\mathcal{D}$ can be defined as follows:

$$\text{DiversityScore} \leftarrow \text{Eval}(\{\tilde{\mathcal{T}}_i\}_{i=1}^n) \qquad (2)$$

where $\text{Eval}(\cdot)$ is the diversity evaluation function, which takes $\{\tilde{\mathcal{T}}_i\}_{i=1}^n$ as inputs and outputs the diversity score.

## 4. Present Work

In this section, we first introduce our proposed method `DCScore` from a classification perspective. Then, we present the properties of `DCScore` followed by theoretical proofs. Finally, we provide a detailed complexity analysis of `DCScore` and the transformation-based counterpart.

### 4.1. `DCScore`: Measuring Diversity from a Classification Perspective

Due to the intrinsic nature of measuring sample differences in diversity evaluation, it is natural to evaluate diversity as a classification task. Consequently, we propose `DCScore`, which formulates diversity evaluation as a sample classification task. Specifically, the difference between samples can be measured through an $n$-classification task, where evaluating $n$ sample datasets involves $n$ $n$-classification tasks, with each sample corresponding to a distinct category. As shown in Figure 1, `DCScore` consists of three stages: text representation, pairwise similarity, and diversity summarization. According to the problem formulation in section 3.2, `DCScore` outputs the diversity of synthetic datasets by evaluating diversity-sensitive components.

Let $\mathcal{D} = \{\mathcal{T}_i\}_{i=1}^n = \{(x_i, y_i)\}_{i=1}^n$ denote a synthetic dataset comprising $n$ input-output samples, and $\{\tilde{\mathcal{T}}_i\}_{i=1}^n$ represents the diversity-sensitive components. `DCScore` adheres to the paradigm of the transformation-based method to evaluate the diversity of $\mathcal{D}$. Specifically, given $\tilde{\mathcal{T}}_i$, `DCScore` first applies an embedding function $\Phi$ to extract the sample representation $\mathbf{h}_i = \Phi(\tilde{\mathcal{T}}_i)$. For all samples in $\mathcal{D}$, we obtain the sample representation matrix $\mathbf{H} \in \mathbb{R}^{n \times d}$ across all samples, where $d$ denotes the dimension of sample representations. Subsequently, `DCScore` utilizes a kernel function $\text{Kernel}(\cdot)$ to calculate a kernel matrix $\mathbf{K}$, where $\mathbf{K} \in \mathbb{R}^{n \times n}$ and entry $\mathbf{K}[i, j]$ represents similarity between $\tilde{\mathcal{T}}_i$ and $\tilde{\mathcal{T}}_j$. From a classification perspective, $\mathbf{K}[i, j]$ can be considered as the logit of $\tilde{\mathcal{T}}_i$ being classified into category $c_j$. where $c_j$ corresponds to $\tilde{\mathcal{T}}_j$. Thus, a sample that is highly distinguishable from others (i.e., correctly classified into its 'own' category with high probability) contributes more to

the overall diversity score. Formally, the aforementioned process can be formulated as follows:

$$\mathbf{H} = \Phi(\{\tilde{\mathcal{T}}_i\}_{i=1}^n), \mathbf{K} = \text{Kernel}(\mathbf{H}), \qquad (3)$$

where $\text{Kernel}(\cdot)$ calculates pairwise similarity, with viable options including inner product and RBF kernel. For $\Phi$, a more expressive embedding function can be employed, such as one trained using a well-designed framework like Sentence-Bert (Reimers, 2019).

Based on $\mathbf{K}$, `DCScore` leverages a classification function with $\mathbf{K}$, denoted as $f_\mathbf{K}(\cdot)$, to compute the classification probability matrix $\mathbf{P} \in \mathbb{R}^{n \times n}$. Here, a natural option for $f_\mathbf{K}(\cdot)$ is the Softmax function. For $\tilde{\mathcal{T}}_i$, the probability that $\tilde{\mathcal{T}}_i$ is classified as category $c_j$ can be formulated as follows:

$$\begin{aligned} P(c = c_j | \tilde{\mathcal{T}}_i) = \mathbf{P}[i, j] &= f_\mathbf{K}(\mathbf{K}[i, j]) \\ &= \frac{\exp{(\mathbf{K}[i, j]/\tau)}}{\sum_j \exp{(\mathbf{K}[i, j]/\tau)}}, \end{aligned} \qquad (4)$$

where $\tau$ is a temperature hyperparameter to control the classification resolution. A smaller $\tau$ amplifies sample similarity differences, implying a higher classification resolution, while a larger value yields the opposite effect.

According to Eq. (4), if the evaluated dataset exhibits high sample richness, indicating greater diversity, each sample is likely to be classified into its own category. If the diversity is low, all samples will be randomly classified. Based on $\mathbf{P}$, `DCScore` calculates diversity of $\mathcal{D}$ as the trace of $\mathbf{P}$, which can be formulated as follows:

**Definition 4.1** (`DCScore`). Let $\mathcal{D} = \{\mathcal{T}_i\}_{i=1}^n$ denote the synthetic dataset with $n$ samples, and let $\{\tilde{\mathcal{T}}_i\}_{i=1}^n$ represent a set of diversity-sensitive components within $\{\mathcal{T}_i\}_{i=1}^n$. Denote $P_i$ as the classification probability vector of $\tilde{\mathcal{T}}_i$. By conducting the classification task for all $\tilde{\mathcal{T}}_i$ and obtaining the probability matrix $\mathbf{P} = [P_1, P_2, ..., P_n]$, `DCScore` for $\mathcal{D}$ is defined as the trace of $\mathbf{P}$:

$$\text{DCScore}(\mathcal{D}) = \text{tr}(\mathbf{P}) = \sum_{i=1}^n \mathbf{P}[i, i]. \qquad (5)$$

*Notably, the process described above is just one implementation of `DCScore`, with other potential implementations to be explored in future work.*

### 4.2. Properties of `DCScore`

We provide theoretical proof that `DCScore` satisfies several axioms (Leinster & Cobbold, 2012) defined for a principled diversity metric. `DCScore` meets four axioms: effective number, identical samples, symmetry, and monotonicity axioms. These axioms ensure a reasonable and robust diversity evaluation. The matched axioms of `DCScore` are outlined below, while their proofs are detailed in Appendix B.

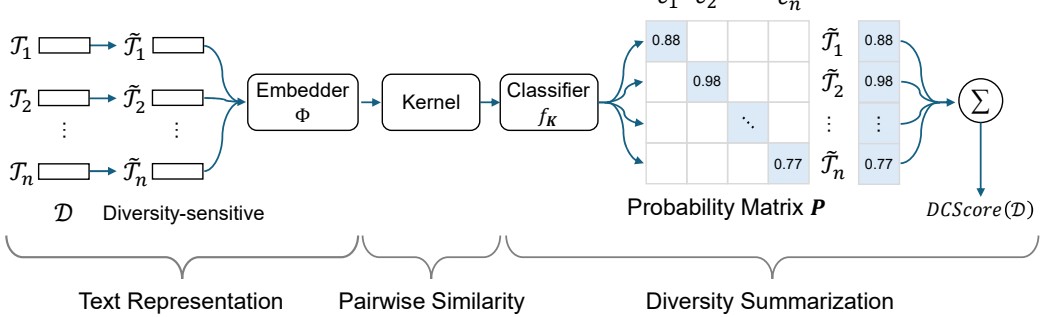

*Figure 1.* The overview of `DCScore`. `DCScore` consists of text representation, pairwise similarity, and diversity summarization stages.

- **Effective number**: Diversity should be defined as the effective number of samples in a dataset, ranging from 1 to $n$. `DCScore` meets this axiom, as evidenced by its behavior: `DCScore` equals 1 when all samples in $\mathcal{D}$ are identical and equals $n$ when all samples are distinct.

- **Identical samples**: Given two identical datasets $\mathcal{D}_1$ and $\mathcal{D}_2$, the diversity of $\mathcal{D}'$ generated by merging these two datasets remains unchanged. The values of `DCScore` are the same across $\mathcal{D}_1$, $\mathcal{D}_2$, and $\mathcal{D}'$, i.e.,

$$\text{DCScore}(\mathcal{D}_1) = \text{DCScore}(\mathcal{D}_2) = \text{DCScore}(\mathcal{D}'). \quad (6)$$

- **Symmetry**: Diversity remains constant regardless of the order of the samples, exhibiting permutation invariance. Let $\pi(\cdot)$ denote the permutation function for the sample order, `DCScore` remains unchanged for any sample permutation of $\mathcal{D}$, i.e.,

$$\text{DCScore}(\mathcal{D}) = \text{DCScore}(\pi(\mathcal{D})). \quad (7)$$

- **Monotonicity**: The diversity of a dataset decreases as the sample similarity increases. Given two datasets $\mathcal{D}_1$ and $\mathcal{D}_2$, and a new sample $\mathcal{T}_{n+1}$, where the samples in $\mathcal{D}_1$ and $\mathcal{D}_2$ are entirely different, and $\text{DCScore}(\mathcal{D}_1) = \text{DCScore}(\mathcal{D}_2) = n$. If $\mathcal{T}_{n+1}$ is more similar to the samples in $\mathcal{D}_2$ than to those in $\mathcal{D}_1$ and is added to both datasets, then for the merged datasets $\mathcal{D}'_1$ and $\mathcal{D}'_2$, `DCScore` satisfies the following equation.

$$\text{DCScore}(\mathcal{D}'_1) > \text{DCScore}(\mathcal{D}'_2). \quad (8)$$

### 4.3. Complexity Analysis

We provide a time complexity analysis of `DCScore` in Table 2. We compare `DCScore` with VendiScore due to their similarity, finding that `DCScore` has lower computational complexity with non-linear kernels.

Denoting $\mathcal{O}_{kernel}$ as the complexity associated with general kernels (i.e., kernels other than linear kernels), we analyze the complexity in the pairwise similarity and summarization

*Table 2.* Complexity analysis of `DCScore` and VendiScore. $\mathcal{O}_{kernel}$ represents the complexity of the kernel function.

| | | General Kernels | Inner Product |
|---|---|---|---|
| **Pairwise Similarity** | VendiScore | $\mathcal{O}(n^2 \cdot \mathcal{O}_{kernel})$ | $\mathcal{O}(d^2 n)$ |
| | `DCScore` | | $\mathcal{O}(n^2 d)$ |
| **Summarization** | VendiScore | $\mathcal{O}(n^3)$ | $\mathcal{O}(d^3)$ |
| | `DCScore` | $\mathcal{O}(n^2)$ | $\mathcal{O}(n^2)$ |
| **Total** | VendiScore | $\mathcal{O}(n^2 \cdot \mathcal{O}_{kernel} + n^3)$ | $\mathcal{O}(d^2 n + d^3) = \mathcal{O}(d^2 n)$ |
| | `DCScore` | $\mathcal{O}(n^2 \cdot \mathcal{O}_{kernel} + n^2)$ | $\mathcal{O}(n^2 d + n^2)$ |

stages. In the pairwise similarity stage, the computation of pairwise similarities results in a complexity of $\mathcal{O}(n^2)$ for `DCScore`. When combined with the complexity $\mathcal{O}_{kernel}$ of the general kernel computation, `DCScore` exhibits a total complexity of $\mathcal{O}(n^2 \cdot \mathcal{O}_{kernel})$. In the summarization stage, `DCScore` has a complexity of $\mathcal{O}(n^2)$ due to the softmax operation. Consequently, the overall complexity of `DCScore` for general kernels is $\mathcal{O}(n^2 \cdot \mathcal{O}_{kernel} + n^2)$. In contrast, VendiScore has a total complexity of $\mathcal{O}(n^2 \cdot \mathcal{O}_{kernel} + n^3)$, where the pairwise similarity stage is identical to that of `DCScore`, while the summarization stage incurs a complexity of $\mathcal{O}(n^3)$ due to the eigenvalue computation. Thus, for general kernels, `DCScore` demonstrates lower complexity than VendiScore.

However, when the inner product is employed as the kernel function and $n \gg d$, VendiScore can significantly reduce the complexity by replacing the pairwise similarity $XX^T$ with $X^T X$, where $X \in \mathbb{R}^{n \times d}$. This results in complexities of $\mathcal{O}(d^2 n)$ for the pairwise similarity stage and $\mathcal{O}(d^3)$ for the summarization stage. In this regard, `DCScore` has a complexity of $\mathcal{O}(n^2 d + n^2)$, which is slightly worse than that of VendiScore. We can leverage efficient techniques, such as those proposed in (Shim et al., 2017), (Milakov & Gimelshein, 2018), and (Wen et al., 2023), to reduce the computational cost of `DCScore`. Compared to VendiScore, `DCScore` maintains lower complexity in most cases, as empirically validated in Section 5.3 and Appendix D.1. While VendiScore has lower complexity with the inner product kernel, experiments in Appendix D.1 indicate that its computation time is similar to `DCScore`. Ensuring low

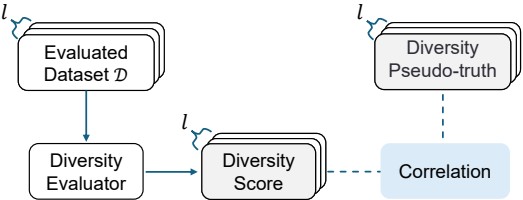

*Figure 2.* Experimental settings of correlation evaluation.

*Table 3.* Correlation (Spearman's $\rho$) between $\tau_g$ and diversity evaluation methods on datasets generated by different settings (*Zero-shot* or *Few-shot*). Spearman's $\rho$ varies between -1 and +1, with 0 implying no correlation. Best results are indicated in **bold**.

| Methods | Zero-shot setting | | | | Few-shot setting | | | |
| | Text classification | | Story completion | | Text classification | | Story completion | |
| | 13B | 70B | 13B | 70B | 13B | 70B | 13B | 70B |
|---|---|---|---|---|---|---|---|---|
| Distinct-n | 0.9909 | **0.9870** | 0.9766 | 0.9701 | 0.9857 | 0.9766 | 0.9779 | 0.9935 |
| K-means Inertia | -0.1143 | 0.9688 | 0.9454 | 0.8727 | 0.7104 | 0.7273 | 0.9662 | 0.9662 |
| VendiScore | **0.9961** | 0.9818 | **0.9870** | **0.9922** | **0.9909** | 0.9857 | **0.9857** | 0.9961 |
| DCScore | **0.9961** | 0.9779 | 0.9844 | 0.9792 | **0.9909** | **0.9883** | **0.9857** | **0.9974** |

computational complexity across multiple kernels is more advantageous than achieving it with a single kernel.

# 5. Experiments

We conduct experiments to verify the effectiveness of DCScore by examining correlation, computational cost, hyperparameter sensitivity, and further probing. Limited by space, we provide additional experiments in Appendix D.

## 5.1. Experimental Settings

To verify the effectiveness of DCScore, we conduct a series of correlation experiments following the setup in (Tevet & Berant, 2020). As shown in Figure 2, the core idea of our experimental evaluation is to correlate the diversity measurement results of DCScore with diversity pseudo-truths, such as the Softmax temperature $\tau_g$ of dataset generation, human judgment, and LLMs evaluation. Specifically, we evaluate $l$ generated datasets to obtain $l$ diversity scores and then calculate the correlation with diversity pseudo-truths. To calculate the correlation between measured diversity scores and diversity pseudo-truths, we employ Spearman's $\rho$ (Spearman, 1961), a measure of rank correlation ranging from -1 to 1, with higher absolute values indicating stronger correlations. Limited by space, we present detailed experimental settings in Appendix C.

**Datasets.** We utilize two categories of datasets in our experiments: self-generated datasets and publicly available generated datasets. Self-generated datasets are generated through two data generation strategies (Li et al., 2023): *zero-shot* and *few-shot* settings. We generate datasets for two natural language processing tasks: *text classification* and *story completion*. Additionally, we utilize three publicly available existing datasets, including SST2 (Socher et al., 2013), Yelp (Zhang et al., 2015), and AG News (Zhang et al., 2015), and their AttrPrompt-augmented version (Yu et al., 2024). Detailed information about these datasets can be found in Appendix C.1.

**Generation Models.** To generate datasets through zero-shot and few-shot settings, we employ two commonly used LLMs as our dataset generators, including Llama2-13B (13B) and Llama2-70B (70B) (Touvron et al., 2023).

**Baseline Methods.** We compare DCScore with three baseline methods detailed in Appendix E, i.e., *Distinct-n* (Li et al., 2015), K-means Inertia (Du & Black, 2019), and VendiScore (Dan Friedman & Dieng, 2023).

## 5.2. Correlation Evaluation

We investigate the correlation between the diversity evaluation of DCScore and diversity pseudo-truths, such as $\tau_g$, human judgment, and LLMs evaluation. We compare DCScore with all baselines on self-generated datasets.

### 5.2.1. CORRELATION WITH $\tau_g$

**Evaluation on self-generated datasets.** Previous works (Caccia et al., 2018; Tevet & Berant, 2020; Chung et al., 2023) have demonstrated a positive correlation between $\tau_g$ and the diversity of generated texts, making $\tau_g$ as a reasonable diversity pseudo-truth. LLMs with lower $\tau_g$ generate less diverse content, whereas higher $\tau_g$ values yield more diverse content. Thus, we evaluate the performance of DCScore on self-generated datasets with varying $\tau_g$, ranging from 0.2 to 1.2 at 0.05 intervals. We present more information about self-generated datasets in Appendix C.1.1. Table 3 displays the correlation results of all methods. All methods accurately capture the true diversity of generated datasets, as demonstrated by high Spearman's $\rho$ values. DCScore performs on par with VendiScore while providing better scalability for larger synthetic datasets, as discussed in Section 5.3. DCScore outperforms all baseline methods under the few-shot setting across all datasets, highlighting its effectiveness. K-means Inertia exhibits the weakest correlation on the text classification dataset generated by the 13B model under the zero-shot setting, potentially due to its sensitivity to the number of cluster centroids. Overall, DCScore outperforms all baselines in most cases, and its evaluation results exhibit a strong correlation with the diversity pseudo-truth according to (Akoglu, 2018).

**Visualization.** We further provide a visualization of the diversity evaluation results for DCScore. For each generated dataset, we prompt LLMs to produce 10 distinct answers corresponding to a single prompt, forming an evaluation batch. We then evaluate diversity using the batch evaluation protocol outlined in Appendix C.2.3.

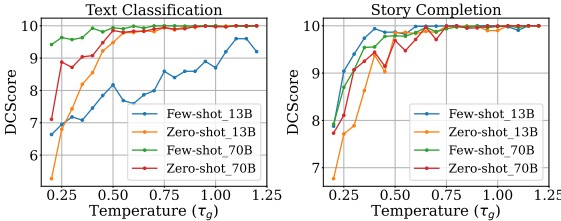

*Figure 3.* Diversity evaluation of `DCScore` on datasets generated using varying $\tau_g$. `DCScore` shows a strong correlation with $\tau_g$, indicating its effectiveness in evaluating the dataset diversity.

Based on the above-mentioned settings, a completely diverse dataset may yield a diversity score of 10. As shown in Figure 3, `DCScore` exhibits a strong positive correlation with $\tau_g$. In most cases, when $\tau_g > 0.75$, `DCScore` scores a generated dataset with a diversity value of approximately 10. For text classification, the 13B generation model under the few-shot setting demonstrates a distinct diversity change pattern compared to others. This phenomenon stems from the inherent limitations of the 13B generation model, which struggles to effectively comprehend and execute more intricate or multi-faceted instructions. As a result, generated datasets only exhibit marginal improvements in diversity.

### 5.2.2. Correlation with Human Judgment

Diversity evaluation is a subjective task, and an ideal method should align well with human judgment. Thus, we investigate the correlation between `DCScore` and human judgment. We enlist three human evaluators to perform pairwise diversity comparisons among datasets with varying $\tau_g$ values and report the diversity ranking by averaging the win rate across evaluators. We conduct all evaluations five times to report average results. Limited by space, we present details of human evaluation in Appendix C.2.5.

Table 4 presents pairwise correlation between human judgment, $\tau_g$, and `DCScore`. Table 4 indicates a strong correlation between human judgment and $\tau_g$, supporting the use of human judgment as a diversity pseudo-truth. Based on this observation, `DCScore` performs better in two settings: **Story-Few** (story completion data generated under the few-shot setting) and **Text-Zero** (text classification data generated under the zero-shot setting). This is confirmed by higher human-`DCScore` correlation in these two settings. For **Story-Zero** and **Text-Few** settings, we observe more identical content in the initial portions of the diversity-sensitive components within an evaluation batch. In these cases, human evaluators tend to disregard the identical content and base their judgments on the latter sections. However, `DCScore` is affected by the identical content, resulting in a lower pairwise correlation. Despite this, the correlation remains strong, as demonstrated by previous studies (Akoglu, 2018).

*Table 4.* Pairwise correlation (Spearman's $\rho$) between human, temperature ($\tau_g$), and `DCScore`. `DCScore` indicates a strong correlation with human judgment.

|  | Story-Few | Story-Zero | Text-Few | Text-Zero |
|---|---|---|---|---|
| Human-`DCScore` | $0.9040_{\pm0.04}$ | $0.7870_{\pm0.10}$ | $0.7915_{\pm0.16}$ | $0.8798_{\pm0.10}$ |
| $\tau_g$-`DCScore` | $0.9086_{\pm0.07}$ | $0.7829_{\pm0.16}$ | $0.8400_{\pm0.16}$ | $0.8971_{\pm0.07}$ |
| $\tau_g$-Human | $0.9276_{\pm0.02}$ | $0.9194_{\pm0.06}$ | $0.9770_{\pm0.02}$ | $0.9255_{\pm0.08}$ |

*Table 5.* Pairwise correlation (Spearman's $\rho$) between GPT-4, temperature ($\tau_g$), and `DCScore`. `DCScore` indicates a strong correlation with GPT-4 evaluation results.

|  | Story-Few | Story-Zero | Text-Few | Text-Zero |
|---|---|---|---|---|
| GPT-4-`DCScore` | $0.6057_{\pm0.30}$ | $0.9010_{\pm0.04}$ | $0.6131_{\pm0.18}$ | $0.9052_{\pm0.09}$ |
| $\tau_g$-`DCScore` | $0.6757_{\pm0.30}$ | $0.8782_{\pm0.08}$ | $0.5714_{\pm0.27}$ | $0.9336_{\pm0.06}$ |
| $\tau_g$-GPT-4 | $0.9086_{\pm0.07}$ | $0.7829_{\pm0.16}$ | $0.8400_{\pm0.16}$ | $0.8971_{\pm0.07}$ |

### 5.2.3. Correlation with LLM Evaluator

To further verify the effectiveness of `DCScore`, we investigate the evaluation correlation between `DCScore` and LLMs. Following the setting in Section 5.2.2, we employ GPT-4 to conduct pairwise comparisons between two generated datasets with different $\tau_g$. These generated datasets are identical to those used in Section 5.2.2. Based on the pairwise comparison results, we obtain the diversity ranking outcomes. Regarding GPT-4 evaluation results as the diversity pseudo-truth, we report the pairwise evaluation correlation between `DCScore`, GPT-4, and $\tau_g$ in Table 5. We observe that `DCScore` exhibits strong correlations with GPT-4 and $\tau_g$ in zero-shot settings. By comparing the results of "$\tau_g$-`DCScore`" and "$\tau_g$-GPT-4", we find that `DCScore` outperforms the GPT-4 evaluator in terms of correlation with $\tau_g$ in zero-shot settings. Regarding the correlation performance in few-shot settings, we notice lower correlations of all baseline methods compared to zero-shot settings. We guess that this phenomenon is related to the distributions of the generated datasets. Although `DCScore` exhibits lower correlations (about 0.6) with GPT-4, this result can still be considered a strong correlation according to (Akoglu, 2018).

*In summary, `DCScore` exhibits a strong correlation (Spearman's $\rho \geq 0.6$), with three diversity pseudo-truths: $\tau_g$, human judgment, and LLMs evaluation, thereby verifying the effectiveness of `DCScore`.*

### 5.3. Computational Cost

The computational cost is crucial in diversity evaluation methods, especially with the increasing sample sizes of synthetic datasets. For a fair comparison, we only present the computation times of transformation-based methods: `DCScore`, K-means Inertia, and VendiScore. We truncate the text length of three datasets (SST2/Yelp/AG News-AttrPrompt, the AttrPrompt (Yu et al., 2024) augmented version of SST2/Yelp/AG News datasets) to 50 tokens and

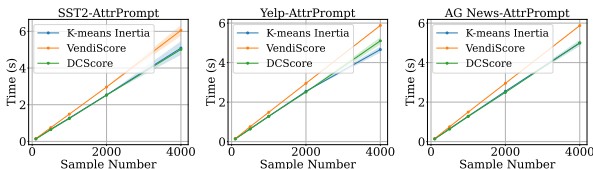

*Figure 4.* Computation times under different sample sizes. `DCScore` outperforms all baselines in computational cost.

*Table 6.* Comparison of computation time between `DCScore` and VendiScore on SST2.

| Kernels | | SST2 | | | | |
|---|---|---|---|---|---|---|
| | Sample num | 4k | 8k | 16k | 32k | 64k |
| Inner product | VendiScore | $4.65_{\pm0.28}$ | $9.84_{\pm0.26}$ | $19.02_{\pm0.70}$ | $37.31_{\pm1.88}$ | $76.19_{\pm1.91}$ |
| | DCScore | $4.58_{\pm0.29}$ | $10.03_{\pm0.17}$ | $20.42_{\pm0.39}$ | $42.91_{\pm1.59}$ | $112.47_{\pm2.43}$ |
| RBF kernel | VendiScore | $5.86_{\pm0.06}$ | $12.41_{\pm0.49}$ | $32.94_{\pm0.40}$ | $100.36_{\pm1.44}$ | $449.14_{\pm10.35}$ |
| | DCScore | $5.22_{\pm0.33}$ | $9.94_{\pm0.42}$ | $21.20_{\pm0.75}$ | $46.57_{\pm1.47}$ | $117.06_{\pm1.91}$ |
| Poly kernel | VendiScore | $5.73_{\pm0.06}$ | $12.72_{\pm0.41}$ | $31.47_{\pm0.97}$ | $98.31_{\pm0.25}$ | $453.11_{\pm2.53}$ |
| | DCScore | $5.09_{\pm0.28}$ | $10.27_{\pm0.12}$ | $20.12_{\pm1.02}$ | $46.25_{\pm1.82}$ | $123.51_{\pm3.40}$ |

record the computation times of three methods with varying sample sizes in the range of $\{100, 500, 1000, 2000, 4000\}$.

As shown in Figure 4, we repeat the experiments five times to report the final results. `DCScore` and K-means Inertia exhibit nearly identical computation times. However, `DCScore` significantly outperforms K-means Inertia in correlation with $\tau_g$, as evidenced in Section 5.2.1. Compared to VendiScore, `DCScore` demonstrates a speed advantage of approximately 16%, or more than one second, when processing 4000 samples. Analyzing the time complexity of these two methods, and disregarding the selection of the kernel function, we find that for a dataset with $n$ samples, where $n \gg d$ is not satisfied, the computational complexity of `DCScore` in diversity summarization is $\mathcal{O}(n^2)$ due to the softmax computation. In contrast, VendiScore requires finding the eigenvalues of an $n \times n$ matrix, resulting in a computational complexity of $\mathcal{O}(n^3)$. Consequently, `DCScore` offers significantly lower time complexity than VendiScore while sacrificing little in diversity evaluation performance. However, as detailed in the complexity analysis shown in Section 4.3, when $n \gg d$ and inner products are used as the kernel function, the total complexity of VendiScore can be reduced to $\mathcal{O}(d^2n)$. Thus, we evaluate computational costs on larger datasets, i.e., satisfying $n \gg d$. As shown in Table 6, `DCScore` exhibits a notable advantage in computational efficiency when using non-linear kernels, e.g., RBF and Poly kernel. We present additional experimental results and a more in-depth analysis in Appendix D.1.

A recent study (Ospanov et al., 2024) employs the random Fourier features framework to decrease the computational cost of entropy-based diversity evaluation methods, such as VendiScore (Dan Friedman & Dieng, 2023) and RKE (Jalali et al., 2023). We follow the experimental settings of Table 6 and leverage different random seeds for data sampling. Limited by space, we present the experimental results in

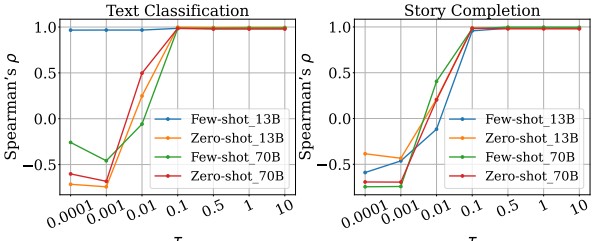

*Figure 5.* Hyperparameter sensitivity analysis w.r.t $\tau$ on self-generated datasets.

Appendix D.2. In summary, `DCScore` demonstrates lower computation time compared to the efficiency-improved versions of VendiScore and RKE in most cases.

### 5.4. Hyperparameter Sensitivity

According to Eq. (4), the temperature ($\tau$) in the Softmax function is a critical hyperparameter that affects classification resolution. To investigate this, we conduct a hyperparameter sensitivity analysis of `DCScore` w.r.t. $\tau$ on self-generated datasets used in Section 5.2.1. We vary $\tau$ within the range of $\{0.0001, 0.001, 0.1, 0.5, 1, 10\}$. Figure 5 presents hyperparameter sensitivity results on datasets for text classification and story completion tasks. Overall, lower $\tau$ values result in lower Spearman's $\rho$, even indicating a negative correlation, while higher $\tau$ values do the opposite. From Eq. (4), a higher $\tau$ reduces pairwise similarity differences, leading to a more uniform distribution of classification probabilities for each sample. This phenomenon can be regarded as a lower classification resolution, i.e., the classification function $f_{\mathbf{K}}$ has poorer discrimination power. Furthermore, the correlation result of the 13B generation model under the few-shot setting for the text classification task remains stable despite variations in $\tau$. This phenomenon has the same explanation as in Figure 3.

### 5.5. Further Probe

#### 5.5.1. DOWNSTREAM TASK TRAINING

To investigate the correlation between `DCScore` and downstream task training, we train text classification models using self-generated datasets under zero-shot and few-shot settings. We vary $\tau_g$ of self-generated datasets within the range of $\{0.2, 0.7, 1.2\}$. More details of training datasets and hyperparameters are presented in Appendix C.1.1 and C.2.2, respectively. As shown in Table 7, models trained on more diverse datasets achieve better accuracy, likely due to their improved generalization capabilities (Gontijo-Lopes et al., 2020). Notably, the increased training data diversity makes model fitting more difficult, necessitating additional epochs to achieve optimal accuracy. Additionally, the diversity evaluated by `DCScore` has a similar trend to the accuracy

*Table 7.* Downstream task training performance and diversity evaluation on self-generated datasets with $\tau_g = \{0.2, 0.7, 1.2\}$.

| | Accuracy | | | DCScore | | |
|---|---|---|---|---|---|---|
| | $\tau_g$=0.2 | $\tau_g$=0.7 | $\tau_g$=1.2 | $\tau_g$=0.2 | $\tau_g$=0.7 | $\tau_g$=1.2 |
| **Zero-shot** | 89.10 | 89.70 | 90.37 | 481.76 | 1745.42 | 2082.42 |
| **Few-shot** | 70.07 | 73.19 | 73.41 | 1376.43 | 1958.16 | 2047.90 |

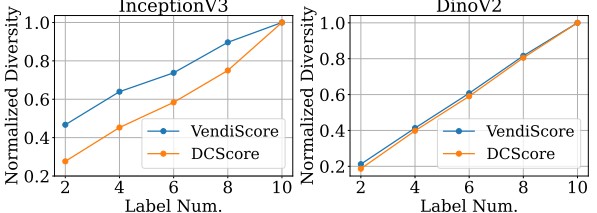

*Figure 6.* Diversity evaluation on colored MNIST using two different embedding functions (Inception V3 and Dino V2). A higher label number signifies greater dataset diversity.

performance in the zero-shot setting, further demonstrating the effectiveness of `DCScore`. Detailed experimental results and further analysis are provided in Appendix D.3.

### 5.5.2. DIVERSITY EVALUATION ON VISUAL MODALITY

Similar to text generation, there is increasing attention on image generation. Thus, we further verify the effectiveness of `DCScore` on the image dataset evaluation. Specifically, following the setting of a recent study (Ospanov et al., 2024), we leverage the colored MNIST (Deng, 2012) as the evaluated dataset and use the label number as the diversity ground truth. Here, a larger label number indicates a higher diversity of the evaluated dataset. We conduct a comparison between `DCScore` and VendiScore, using inner product as the kernel function, while adopting Inception V3 and Dino V2 as the embedding functions. For `DCScore`, $\tau$ is set to 1. Figure 6 illustrates the diversity evaluation results on the image dataset, with diversity results normalized by dividing by the highest result of each method. We observe that both `DCScore` and VendiScore exhibit a positive correlation with the number of labels. When using the Inception V3 model as the embedding function, `DCScore` demonstrates higher correlations with the number of labels compared to VendiScore, as evidenced by a line that is closer to the diagonal. Additionally, `DCScore` presents more consistent results across different embedding functions compared to VendiScore, indicating stable performance with respect to the embedding function.

### 5.5.3. DIVERSITY EVALUATION ON DATASETS WITH DUPLICATE SAMPLES

To further investigate the limitations of `DCScore`, we conduct diversity evaluations on datasets with duplicate sam-

*Table 8.* Comparison of evaluation stability between `DCScore` and VendiScore on datasets with duplicate samples. $n$ denotes the number of duplicate samples, and A.P. represents AttrPrompt.

| Datasets | Methods | $n$=0 | $n$=10 | $n$=100 | $n$=1000 | $n$=3000 | $n$=6000 |
|---|---|---|---|---|---|---|---|
| SST2 A.P. | DCScore | 5937.00 | 5936.99 | 5937.10 | 5936.81 | 5936.73 | 5936.99 |
| | VendiScore | 11.31 | 11.31 | 11.31 | 11.32 | 11.29 | 11.31 |
| Yelp A.P. | DCScore | 5939.94 | 5940.01 | 5939.82 | 5939.91 | 5940.21 | 5939.94 |
| | VendiScore | 8.22 | 8.22 | 8.22 | 8.24 | 8.24 | 8.22 |
| AG News A.P. | DCScore | 5998.04 | 5998.04 | 5998.04 | 5998.04 | 5997.75 | 5998.04 |
| | VendiScore | 40.71 | 40.70 | 40.70 | 40.63 | 40.69 | 40.71 |

ples. Specifically, we randomly select $n$ samples from the evaluated datasets to serve as duplicates and incorporate them into the original datasets. In our experiments, we use SST2-AttrPrompt, Yelp-AttrPrompt, and AG News-AttrPrompt as evaluated datasets, setting $n$ to $\{0, 10, 100, 1000, 3000, 6000\}$. Table 8 presents the diversity evaluation values of `DCScore` and VendiScore, using inner product as the kernel function. These values can be interpreted as the number of effective samples. As shown in Table 8, we observe that both `DCScore` and VendiScore maintain stable evaluation results as the number of duplicate samples increases. This indicates that both methods are robust in terms of the impact of duplicates. Additionally, the results for VendiScore tend to be lower, suggesting an underestimation of dataset diversity.

## 6. Conclusion

In this work, we investigate the diversity evaluation of synthetic datasets, a topic systematically under-explored in existing research. To this end, we present `DCScore`, a diversity evaluation method from a classification perspective. `DCScore` regards the holistic diversity evaluation as the classification task at the sample level, thereby facilitating the capture of mutual relationships between samples. We provide theoretical guarantees demonstrating that `DCScore` meets the axiom requirements (Leinster & Cobbold, 2012) for a principled diversity evaluation method. Experiments on synthetic datasets reveal that `DCScore` exhibits better correlations with various diversity pseudo-truths, including $\tau_g$, human judgment, and LLMs evaluation. Meanwhile, `DCScore` exhibits significantly lower computational cost compared to transformation-based counterparts. Finally, we hope our work encourages future research to pay more attention to the diversity of synthetic datasets and promotes the wider application of these datasets.

**Limitation:** Although we have verified the effectiveness of `DCScore` for unimodal data evaluation, `DCScore` is not yet capable of directly evaluating multimodal data, such as image-text pairs. A key challenge lies in the extraction and fusion of representations from multimodal data. As multimodal LLMs continue to enhance their generative capabilities, evaluating the diversity of multimodal-generated datasets emerges as an important research problem.

## Acknowledgements

The research is supported by the National Key R&D Program of China under grant No. 2022YFF0902500, the Guangdong Basic and Applied Basic Research Foundation, China (No. 2023A1515011050), and Tencent AI Lab RBFR2024004.

## Impact Statement

This paper presents a novel and efficient method for measuring the diversity of synthetic datasets, aiming to advance fields of machine learning and LLMs. The ability to accurately evaluate the diversity of synthetic data has significant implications for various applications, including but not limited to improving the robustness and fairness of ML models, enhancing the quality of synthetic data used in training, and fostering innovation in generative models.

From an ethical perspective, our work contributes to the responsible development of AI by providing tools that can help identify and mitigate biases in synthetic data. Ensuring diversity in datasets is crucial for creating equitable AI systems that perform well across different demographic groups and use cases. By promoting diversity, our method can help prevent the perpetuation of existing biases and support the creation of more inclusive technologies.

For societal impact, our approach has the potential to impact a wide range of industries, from healthcare to finance, by enabling the generation of more representative and diverse datasets. This can lead to more accurate and fair decision-making processes, ultimately benefiting society as a whole.

Overall, while our primary goal is to advance fields of machine learning and LLMs, we believe that our work also addresses important ethical considerations and has the potential to contribute positively to society by promoting fairness and inclusivity in AI systems.

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

# A. Additional Related Work

Limited by space, we provide a literature review of the LLM dataset generator and application of diversity evaluation methods as follows.

## A.1. LLM Dataset Generator

Recent studies (Ding et al., 2022; Chung et al., 2023) leverage LLMs to augment existing datasets or generate a dataset from scratch, demonstrating the effectiveness in improving dataset quality and reducing data collection costs. Generally, efforts to employ LLMs as dataset generators can be categorized into three strategies: *Prompt-guided* (Li et al., 2023), *Dataset-guided* (Ye et al., 2022), and *Instruct-guided* (Samuel et al., 2023).

**Prompt-guided and Dataset-guided Strategies.** The prompt-guided strategy, a prominent data augmentation approach using LLMs, involves designing task-specific prompts to guide LLMs to augment data in a few-shot (Yoo et al., 2021) or zero-shot (Mahmoudi et al., 2024) manner. Due to its simplicity and effectiveness, subsequent works extend this strategy to various scenarios, such as medical (Yuan et al., 2023), person retrieval (Li et al., 2024b), and social media scenario (dos Santos et al., 2024). However, simple prompt engineering has limitations in fully exploiting the capabilities of LLMs, leading to the development of multi-level prompt designs (Ye et al., 2024) and targeted sample augmentation (Yang et al., 2024b). To further harness the potential of LLMs, the dataset-guided strategy employs LLMs to generate a training set and then trains a task-specific model to annotate unlabeled data (Sahu & Laradji, 2024). The dataset-guided strategy aims to approximate the distribution of targeted scenarios, but it is currently only applicable to text classification tasks.

**Instruct-guided Strategy.** Previous studies (White et al., 2023) indicate that the design of prompts significantly impacts the performance of LLMs, spurring research into the instruct-guided strategy. Generally speaking, the instruct-guided strategy leverages LLMs to generate instructions that guide another LLM in dataset generation (Evuru et al., 2024). These instructions typically relate to context (Samuel et al., 2023), criteria (Huang et al., 2023), and tasks (Wang et al., 2022). To further improve the quality of instructions, efforts have been concentrated on selecting optimal instructions (Li et al., 2024a), integrating soft instructions (Chen et al., 2023), and implementing self-correction mechanisms (Gupta et al., 2023).

In a nutshell, LLMs are employed to generate or augment datasets through prompt engineering and multi-step strategies, which encompass various application scenarios and downstream tasks. Meanwhile, the diversity of synthetic datasets emerges as a critical factor in measuring data quality. In our work, we focus on the diversity evaluation of synthetic datasets derived from any dataset generation strategies.

## A.2. Application of Diversity Evaluation Methods

Extending beyond diversity quantification, these methods demonstrate wider utility, such as quantifying augmentation performance and evaluating mode collapse. Thus, we present a literature review of other applications of the diversity evaluation.

**Quantifying Augmentation Performance.** As data augmentation becomes an essential component in the training of deep neural networks (Zhang, 2017; Park et al., 2019), researchers gradually explore a better quantification of the quality of data augmentation. Some studies (Cubuk et al., 2019) suggest that the effectiveness of data augmentation arises from the increased diversity of the data. Inspired by this observation, a series of studies have introduced diversity evaluation metrics into the performance assessment of data augmentation strategies. Specifically, they consider diversity as one aspect of evaluating the quality of augmented data, thereby determining the effectiveness of data augmentation. For instance, (Gontijo-Lopes et al., 2020) utilizes the fundamental idea that models find it more challenging to fit more diverse data, comparing metrics such as training loss and training time before and after augmentation to assess diversity. Similarly, (Yang et al., 2024a) evaluates diversity by examining the eigenvalues and eigenvectors of the similarity matrix of samples before and after augmentation.

**Evaluating Mode Collapse.** Generative adversarial networks (GANs) (Goodfellow et al., 2020) suffer from a well-known phenomenon called mode collapse, which can result in a lack of diversity in the generated samples (Dieng et al., 2019). Consequently, existing studies assess mode collapse by evaluating the diversity of the generated samples. For instance, a common approach is to train an MNIST classifier and then count the number of unique classes predicted for the generated samples. Following this paradigm, VendiScore (Dan Friedman & Dieng, 2023) compares the generation diversity of PresGAN (Dieng et al., 2019) and Self-conditioned GAN (Liu et al., 2020). Additionally, some studies (Yu et al., 2017; Zhu et al., 2018; Caccia et al., 2018) employ different metrics to evaluate the diversity of generated samples from GANs.

**Other Applications.** In addition to the aforementioned applications, diversity evaluation metrics have valuable applications in various areas, including sample selection for datasets (Cao et al., 2023), enhancing adversarial robustness (Lee et al., 2022) and out of distribution robustness (Leng et al., 2025), and eliminating biases within datasets (Huber et al., 2024).

# B. Proof of Properties of `DCScore`

We theoretically confirm that `DCScore` satisfies several intuitive axioms pointed out by previous studies (Leinster & Cobbold, 2012), thereby demonstrating its role as a principled diversity evaluation method.

- **Effective number (Restated)**: Diversity should be defined as the effective number of samples in a dataset, ranging from 1 to $n$. `DCScore` meets this axiom, as evidenced by its behavior: `DCScore` equals 1 when all samples in $\mathcal{D}$ are identical and equals $n$ when all samples are distinct.

  *Proof.* For `DCScore`, if all samples in a dataset are the same, the probability of any given sample being classified into all categories is the same, i.e., for all $i, j = \{1, 2, ..., n\}$, $\mathbf{P}[i, i] = \mathbf{P}[i, j] = \frac{1}{n}$. Then, we have DCScore $= \sum_{i=1}^{n} \frac{1}{n} = 1$. If all samples in the dataset are distinct, for all $i, j = \{1, 2, ..., n\}$, $\mathbf{P}[i, i] = 1$. In other words, the classification function confidently predicts that $\tilde{\mathcal{T}}_i$ belongs to the $i$-th category. Then, we have DCScore tending to $n$. $\square$

- **Identical samples (Restated)**: Given two identical datasets $\mathcal{D}_1$ and $\mathcal{D}_2$, the diversity of the synthetic dataset $\mathcal{D}'$ generated by merging these two datasets remains unchanged. The values of `DCScore` are the same across $\mathcal{D}_1$, $\mathcal{D}_2$, and $\mathcal{D}'$, i.e.,

$$\mathrm{DCScore}(\mathcal{D}_1) = \mathrm{DCScore}(\mathcal{D}_2) = \mathrm{DCScore}(\mathcal{D}'). \tag{9}$$

  *Proof.* Assuming that $\mathcal{D}_1$ and $\mathcal{D}_2$ are completely identical, and the samples within each dataset are entirely different, i.e., $\mathrm{DCScore}(\mathcal{D}_1) = \mathrm{DCScore}(\mathcal{D}_2) = n$. Let $\mathbf{P} = [P_1, ..., P_n, ..., P_{2n}]$ denote the probability matrix of the merged dataset $\mathcal{D}' = \mathcal{D}_1 \cup \mathcal{D}_2 = \{\mathcal{T}_i\}_{i=1}^{2n}$. For $1 \le i \le n$, $\mathcal{T}_i = \mathcal{T}_{n+i}$, where $\mathcal{T}_i \in \mathcal{D}_1$, $\mathcal{T}_{n+i} \in \mathcal{D}_2$. Consequently, for each diversity-sensitive component $\tilde{\mathcal{T}}_i$ in $\mathcal{D}'$, $\mathbf{P}[i, i] = \mathbf{P}[i, n+i] = \frac{1}{2}$. Finally, $\mathrm{DCScore}(\mathcal{D}') = \sum_{i=1}^{2n} \frac{1}{2} = n$.

  However, the assumption that all samples in the dataset are completely different may be too stringent. We further provide a proof with a more relaxed assumption. Suppose that $\mathcal{D}_1$ and $\mathcal{D}_2$ are completely identical, with $\mathbf{K}_1$ and $\mathbf{K}_2$ denoting the kernel matrices for $\mathcal{D}_1$ and $\mathcal{D}_2$, respectively. In this case, we have $\mathbf{K}_1 = \mathbf{K}_2$ as follows:

$$\mathbf{K}_1 = \mathbf{K}_2 = \begin{bmatrix} k_{1,1}^1 & k_{1,2}^1 & \cdots & k_{1,n}^1 \\ k_{2,1}^1 & k_{2,2}^1 & \cdots & k_{2,n}^1 \\ \vdots & \vdots & \ddots & \vdots \\ k_{n,1}^1 & k_{n,2}^1 & \cdots & k_{n,n}^1 \end{bmatrix} = \begin{bmatrix} k_{1,1}^2 & k_{1,2}^2 & \cdots & k_{1,n}^2 \\ k_{2,1}^2 & k_{2,2}^2 & \cdots & k_{2,n}^2 \\ \vdots & \vdots & \ddots & \vdots \\ k_{n,1}^2 & k_{n,2}^2 & \cdots & k_{n,n}^2 \end{bmatrix}. \tag{10}$$

According to Eq. (4), for the $i$-th diversity-sensitive component in $\mathcal{D}_1$ and $\mathcal{D}_2$, the probability of being classified as category $c_i$ can be computed as follows:

$$\mathbf{P}_1[i, i] = \mathbf{P}_2[i, i] = \frac{k_{i,i}^1}{\sum_j k_{i,j}^1} = \frac{k_{i,i}^2}{\sum_j k_{i,j}^2}. \tag{11}$$

For a merged dataset $\mathcal{D}' = \mathcal{D}_1 \cup \mathcal{D}_2 = \{\mathcal{T}_i\}_{i=1}^{2n}$, when $1 \le i \le n$, we have $\mathcal{T}_i = \mathcal{T}_{n+i}$, where $\mathcal{T}_i \in \mathcal{D}_1$, $\mathcal{T}_{n+i} \in \mathcal{D}_2$. Since the newly added data samples do not affect the pairwise similarity, the kernel matrix $\mathbf{K}'$ for $\mathcal{D}'$ can be formulated as follows:

$$\mathbf{K}' = \begin{bmatrix} k_{1,1}^1 & \cdots & k_{1,n}^1 & k_{1,1}^2 & \cdots & k_{1,n}^2 \\ \vdots & \ddots & \vdots & \vdots & \ddots & \vdots \\ k_{n,1}^1 & \cdots & k_{n,n}^1 & k_{n,1}^2 & \cdots & k_{n,n}^2 \\ k_{1,1}^2 & \cdots & k_{1,n}^2 & k_{1,1}^1 & \cdots & k_{1,n}^1 \\ \vdots & \ddots & \vdots & \vdots & \ddots & \vdots \\ k_{n,1}^2 & \cdots & k_{n,n}^2 & k_{n,1}^1 & \cdots & k_{n,n}^1 \end{bmatrix}. \tag{12}$$

Analogous to Eq. (11), for $1 \leq i \leq n$, the probability of the $i$-th diversity-sensitive component in $\mathcal{D}'$ being classified as category $c_i$ can be computed as follows:

$$
\begin{aligned}
\mathbf{P}'[i,i] &= \frac{k_{i,i}^1}{\sum_j k_{i,j}^1 + \sum_j k_{i,j}^2} \\
&= \frac{k_{i,i}^1}{2\sum_j k_{i,j}^1} = \frac{k_{i,i}^1}{2\sum_j k_{i,j}^2} \\
&= \frac{1}{2}\mathbf{P}_1[i,i] = \frac{1}{2}\mathbf{P}_2[i,i].
\end{aligned}
\tag{13}
$$

For $n+1 \leq i \leq 2n$, we obtain the same result as depicted in Eq. (13). Consequently, the diversity of $\mathcal{D}'$ can be computed as follows:

$$
\begin{aligned}
\text{DCScore}(\mathcal{D}') &= \sum_i^{2n} \mathbf{P}'[i,i] \\
&= \frac{1}{2}\sum_i^n \mathbf{P}_1[i,i] + \frac{1}{2}\sum_i^n \mathbf{P}_2[i,i] \\
&= \sum_i^n \mathbf{P}_1[i,i] = \sum_i^n \mathbf{P}_2[i,i] \\
&= \text{DCScore}(\mathcal{D}_1) = \text{DCScore}(\mathcal{D}_2).
\end{aligned}
\tag{14}
$$

$\square$

- **Symmetry (Restated)**: Diversity remains constant regardless of the order of the samples, exhibiting permutation invariance. Let $\pi(\cdot)$ denote the permutation function for the sample order, `DCScore` remains unchanged for any sample permutation of $\mathcal{D}$, i.e.,

$$
\text{DCScore}(\mathcal{D}) = \text{DCScore}(\pi(\mathcal{D})).
\tag{15}
$$

*Proof.* According to Eq. (4), the order of samples does not affect the classification task. Thus, the diagonal elements of $\mathbf{P}$ remain unchanged, indicating the symmetry property of `DCScore`. $\square$

- **Monotonicity (Restated)**: The diversity of a dataset decreases as the sample similarity increases. Given two datasets $\mathcal{D}_1$ and $\mathcal{D}_2$, and a new sample $\mathcal{T}_{n+1}$, where the samples in $\mathcal{D}_1$ and $\mathcal{D}_2$ are entirely different, and $\text{DCScore}(\mathcal{D}_1) = \text{DCScore}(\mathcal{D}_2) = n$. If $\mathcal{T}_{n+1}$ is more similar to the samples in $\mathcal{D}_2$ than to those in $\mathcal{D}_1$ and is added to both datasets, then for the merged datasets $\mathcal{D}_1'$ and $\mathcal{D}_2'$, `DCScore` satisfies the following equation.

$$
\text{DCScore}(\mathcal{D}_1') > \text{DCScore}(\mathcal{D}_2').
\tag{16}
$$

*Proof.* For $\mathcal{D}_1' = \{\mathcal{T}_1^1, \mathcal{T}_2^1, ..., \mathcal{T}_n^1, \mathcal{T}_{n+1}\}$ and $\mathcal{D}_2' = \{\mathcal{T}_1^2, \mathcal{T}_2^2, ..., \mathcal{T}_n^2, \mathcal{T}_{n+1}\}$, we have $\text{S}(\mathcal{T}_i^1, \mathcal{T}_{n+1}) < \text{S}(\mathcal{T}_j^2, \mathcal{T}_{n+1})$ for any $i, j = \{1, 2, ..., n\}$. Here, $\text{S}(\cdot, \cdot)$ is the similarity function. In this regard, the classification function $f(\cdot)$ exhibits lower confidence when classifying dataset $\mathcal{D}_2'$, resulting in a lower probability that the $i$-th sample is classified into the $i$-th class, thereby leading to $\mathbf{P}_{\mathcal{D}_1'}[i,i] > \mathbf{P}_{\mathcal{D}_2'}[i,i]$. Then, the following formula is satisfied:

$$
\mathbf{P}_{\mathcal{D}_1'}[i,i] > \mathbf{P}_{\mathcal{D}_2'}[i,i] \rightarrow \text{DCScore}(\mathcal{D}_1') > \text{DCScore}(\mathcal{D}_2'),
\tag{17}
$$

where $\mathbf{P}_{\mathcal{D}_1'}, \mathbf{P}_{\mathcal{D}_2'}$ are the probability matrix of $\mathcal{D}_1', \mathcal{D}_2'$, respectively. $\square$

## C. Experimental Settings

### C.1. Datasets

Two types of generated datasets, including self-generated datasets and publicly available generated datasets, are employed in our experiments. We provide detailed information on these datasets below.

### C.1.1. SELF-GENERATED DATASETS

In our experiments, we utilize three different self-generated datasets. We employ two commonly used LLMs as our dataset generator, including Llama2-13B (13B) and Llama2-70B (70B) (Touvron et al., 2023). To prompt LLMs to generate datasets, we design two prompts corresponding to *Zero-shot* and *Few-shot* generation settings, respectively. Additionally, self-generated datasets involve two natural language processing tasks: *text classification* and *story completion*. We set the maximum number of generated tokens to 100 and 30 for text classification and story completion tasks, respectively. The detailed generation information is offered as follows.

**Generation Settings.** We use three different generation settings across different experiments. The detailed information is shown as follows.

- **Datasets on Section 5.2.1, Section 5.4, Appendix D.4, and Appendix D.5.** We generate 21 sub-datasets corresponding to different $\tau_g$ by varying $\tau_g$ from 0.2 to 1.2 with 0.05 intervals. For each sub-dataset, we employ LLMs (13B or 70B) to generate sets of 10 responses per context. Specifically, each sub-dataset consists of 100 samples.

- **Datasets on Section 5.2.2 and Section 5.2.3.** We employ the 70B model to generate 6 sub-datasets corresponding to different $\tau_g$ by varying $\tau_g$ from 0.2 to 1.2 with 0.2 intervals. Each sub-dataset includes 5 samples corresponding to a context. To repeat experiments five times, we use five different contexts to prompt the 70B model to generate 5 sub-datasets for each $\tau_g$.

- **Datasets on Appendix 5.5.1 and Appendix D.3.** In zero-shot or few-shot settings, we utilize the 70B model to generate three sub-datasets for the text classification task, corresponding to $\tau_g = \{0.2, 0.7, 1.2\}$, respectively. Unlike other settings that provide only one example, in this experiment, we adopt a few-shot setting where four examples and their corresponding labels are given, including two positive examples and two negative examples. Each sub-dataset contains 3,000 samples, and a context is employed to prompt the 70B model to generate five samples. To train text classification models on each sub-dataset, we randomly split 2,100 samples to the training set for each sub-dataset and gather the remaining 900 samples into the testing set across all three sub-datasets. Consequently, we construct a test set comprising 1,800 samples.

**Prompt Settings.** Following the setting of (Li et al., 2023), we design different prompts for zero-shot and few-shot settings, respectively. Here, we provide a seed example for the few-shot setting, whereas the zero-shot setting does not receive any. For the text classification task under the zero-shot setting, we require LLMs to generate movie reviews with Sci-fi/Action/Drama/Comedy/Romance topics. Each movie review contains a single sentiment of either positive or negative, which is regarded as the text label. For the story completion task, we require LLMs to complete the story according to the given context. The detailed prompt setting is provided in Table 9.

In Table 9, "{style}" will be replaced with one topic within {Sci-fi, Action, Drama, Comedy, Romance} and "{pos_or_neg}" will be replaced with one label within {Positive, Negative}. "{num_of_words}" will be replaced with "50". "{story_q}" will be replaced by the first three sentences of each sample in the ROC Stories dataset.

### C.1.2. PUBLICLY AVAILABLE GENERATED DATASETS

We use SST2 (Socher et al., 2013), Yelp (Zhang et al., 2015), and AG News (Zhang et al., 2015), and their augmented version based on AttrPrompt (Yu et al., 2024). For three original datasets, we randomly sample data from training sets and apply this to the computational cost analysis in Section 5.3, Appendix D.1, and Appendix D.2. For three augmented datasets, each dataset has 6000 samples. We sample different sub-datasets based on these three datasets, applied to Section 5.3 and Section 5.5.3, respectively. The details are as follows.

- **Datasets on Section 5.3.** We remove samples with text token lengths less than 50 in the three datasets and then truncate each sample to a length of 50 tokens. Based on the above, we set up sub-datasets with randomly selected samples of 100, 500, 1000, 2000, and 4000.

### C.2. Implementation Details

For three transformation-based methods, including `DCScore`, VendiScore, and K-means Inertia, we employ *unsup-simcse-bert-base-uncased* (princeton nlp, 2021) as the weight of the embedding function. For all language models used to generate

*Table 9.* Prompt settings for *zero-shot* and *few-shot* settings. Contents that need to be replaced are highlighted in gray.

| NLG Tasks | Zero-shot | Few-shot |
|---|---|---|
| **Text Classification** | Now you are a movie critic. You are given a movie genre/style and a length requirement. You must come up with a movie that corresponds to the genre/style and write a review that meets the length requirement. Write a film review for a {style} movie to express {pos_or_neg} feedback. Each review should have {num_of_words} words. Be sure to express your personal insights and feelings. Please be creative and write unique movie reviews. | Now you are a movie critic. You are given a movie genre/style and a length requirement. You must come up with a movie that corresponds to the genre/style and write a review that meets the length requirement. Write a film review according to the given example. Make sure your review expresses the same sentiment (positive or negative) as the example. Each review should have {num_of_words} words. Be sure to express your personal insights and feelings. Please be creative and write unique movie reviews. The following is the example: #An example from IMDB (Maas et al., 2011)# |
| **Story Completion** | Question: {story_q} 
 Answer: | Complete the story according to the given example. 
 Example: 
 #An example from ROC Stories (Mostafazadeh et al., 2016)# 
 Question: {story_q} 
 Answer: |

the dataset, we set the top-p and top-k parameters to 1 and -1, respectively. Additionally, we limit the maximum number of newly generated tokens to 100 for the text classification task and 30 for the story completion task. All experiments are conducted on $8\times$ NVIDIA Tesla V100 GPU with 32GB of memory. For self-generated datasets, we only evaluate the diversity of generated components.

### C.2.1. HYPERPARAMETER SETTINGS OF DIVERSITY EVALUATION

For `DCScore`, VendiScore, and K-means Inertia, we fix the batch size of generating sample representation at 128 across all experiments. Given the varying hyperparameters for each diversity evaluation method, we provide the detailed settings for each method below:

- **DCScore.** We employ the inner product as $\text{Kernel}(\cdot)$, and Softmax as $f_{\mathbf{K}}(\cdot)$. Except for hyperparameter sensitivity experiments, we set $\tau$ in Eq. (4) to 1 for all other experiments.

- **Distinct-n.** We use 5-grams to calculate distinct-n.

- **K-means Inertia.** We set the number of clusters to 10 for all experiments.

- **VendiScore.** We employ the inner product as $\text{Kernel}(\cdot)$.

### C.2.2. HYPERPARAMETER SETTINGS OF DOWNSTREAM TASK TRAINING

To train text classification models, we employ RoBERTa (Liu, 2019) as the encoder and utilize the representations from the last layer of the encoder as the classifier's input. We employ LoRA (Hu et al., 2021) to finetune the encoder and the

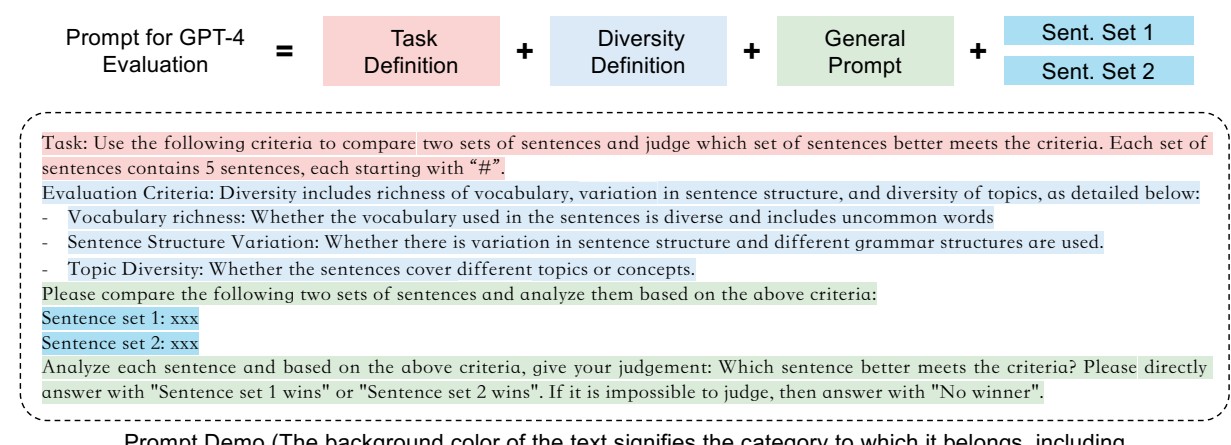

Prompt Demo (The background color of the text signifies the category to which it belongs, including the task definition, diversity definition, general prompt, and sentence sets.)

*Figure 7.* Prompt settings for GPT-4 evaluations.

classifier on self-generated datasets. Specifically, we fix the LoRA scaling factor to 32 and the rank of the update matrices to 8. We use AdamW (Loshchilov, 2017) with an initial learning rate of 5e-5 and linear learning rate decay as our optimizer. Additionally, we set the batch size per GPU as 32 and epochs as 120. For the number of training GPUs, we employ 8 GPUs for zero-shot settings and 4 GPUs for few-shot settings. Therefore, the different training steps for zero-shot and few-shot settings are shown in Figure 8.

### C.2.3. EVALUATION PROTOCOL

In our experiments, we employ diversity evaluation methods to score the diversity of sub-datasets using two evaluation protocols: *overall evaluation* and *batch evaluation*. While K-means Inertia uses the overall evaluation protocol, all other methods utilize the batch evaluation protocol. The detailed settings for the two evaluation protocols are as follows:

- **Batch evaluation.** Due to a context or prompt associated with several samples in a sub-dataset, the batch evaluation protocol requires that evaluation methods treat samples generated from the same context as a single batch. The evaluation results are then averaged across all batches of the entire sub-dataset.

- **Overall evaluation.** We consider each sample in a sub-dataset as independent, meaning each sample is generated by a distinct context or prompt. Based on this assumption, the overall evaluation protocol requires evaluation methods to directly measure the diversity of the entire sub-dataset.

### C.2.4. PROMPT SETTINGS OF LLM EVALUATION

In Section 5.2.3, we use GPT-4 to perform pairwise diversity comparisons. To guide GPT-4 in making these comparisons, we employ a well-designed prompt, as illustrated in Figure 7. The prompt for GPT-4 evaluations includes the task definition, diversity definition, general prompt, and sentence sets to be compared.

### C.2.5. SETTINGS OF HUMAN EVALUATION

According to Appendix C.1.1, datasets in Section 5.2.2 are generated at six temperatures, with five results per context (prompt) in each sub-dataset. During the evaluation, evaluators are asked to select the more diverse sub-dataset from pairs of sub-datasets. Across six temperatures, this results in 15 comparisons, and with three evaluators, a total of 45 judgments are made. Sub-datasets are ranked by the frequency of being chosen as more diverse. This process is repeated five times with different contexts to derive the final human diversity ranking.

*Table 10.* Comparison of computation time between `DCScore` and VendiScore on Yelp.

| Kernels | Yelp | | | | | |
|---|---|---|---|---|---|---|
| | Sample num | 4k | 8k | 16k | 32k | 64k |
| Inner product | VendiScore | $57.96_{\pm 0.35}$ | $\mathbf{114.64}_{\pm 1.63}$ | $\mathbf{227.76}_{\pm 7.04}$ | $\mathbf{451.49}_{\pm 19.73}$ | $\mathbf{912.60}_{\pm 25.69}$ |
| | `DCScore` | $\mathbf{57.95}_{\pm 0.31}$ | $115.35_{\pm 1.16}$ | $232.49_{\pm 1.34}$ | $448.98_{\pm 23.94}$ | $961.29_{\pm 2.86}$ |
| RBF kernel | VendiScore | $59.31_{\pm 0.06}$ | $118.15_{\pm 0.91}$ | $242.06_{\pm 7.60}$ | $527.99_{\pm 2.89}$ | $1272.93_{\pm 21.15}$ |
| | `DCScore` | $\mathbf{58.49}_{\pm 0.14}$ | $\mathbf{116.29}_{\pm 0.92}$ | $\mathbf{232.94}_{\pm 3.09}$ | $\mathbf{471.18}_{\pm 7.80}$ | $\mathbf{953.62}_{\pm 17.21}$ |
| Poly kernel | VendiScore | $59.48_{\pm 0.05}$ | $118.94_{\pm 0.95}$ | $234.08_{\pm 11.72}$ | $522.82_{\pm 3.04}$ | $1313.55_{\pm 12.64}$ |
| | `DCScore` | $\mathbf{58.73}_{\pm 0.08}$ | $\mathbf{117.02}_{\pm 0.90}$ | $\mathbf{227.72}_{\pm 9.51}$ | $\mathbf{462.45}_{\pm 13.91}$ | $\mathbf{988.53}_{\pm 1.10}$ |

*Table 11.* Comparison of computation time between `DCScore` and VendiScore on AG News.

| Kernels | AG News | | | | | |
|---|---|---|---|---|---|---|
| | Sample num | 4k | 8k | 16k | 32k | 64k |
| Inner product | VendiScore | $\mathbf{14.56}_{\pm 1.16}$ | $\mathbf{30.20}_{\pm 1.15}$ | $63.70_{\pm 1.39}$ | $\mathbf{127.25}_{\pm 1.13}$ | $\mathbf{254.20}_{\pm 11.71}$ |
| | `DCScore` | $14.61_{\pm 1.15}$ | $30.61_{\pm 1.77}$ | $\mathbf{63.57}_{\pm 2.68}$ | $129.70_{\pm 4.17}$ | $284.76_{\pm 12.30}$ |
| RBF kernel | VendiScore | $16.69_{\pm 1.54}$ | $33.69_{\pm 1.47}$ | $80.09_{\pm 2.34}$ | $185.79_{\pm 6.44}$ | $617.06_{\pm 12.51}$ |
| | `DCScore` | $\mathbf{16.01}_{\pm 1.53}$ | $\mathbf{31.06}_{\pm 0.96}$ | $\mathbf{69.15}_{\pm 1.32}$ | $\mathbf{129.36}_{\pm 5.56}$ | $\mathbf{297.29}_{\pm 3.67}$ |
| Poly kernel | VendiScore | $17.60_{\pm 0.62}$ | $36.16_{\pm 1.27}$ | $79.34_{\pm 1.57}$ | $190.96_{\pm 2.75}$ | $632.69_{\pm 10.14}$ |
| | `DCScore` | $\mathbf{16.88}_{\pm 0.59}$ | $\mathbf{33.78}_{\pm 1.28}$ | $\mathbf{68.18}_{\pm 1.66}$ | $\mathbf{138.18}_{\pm 3.82}$ | $\mathbf{303.06}_{\pm 11.40}$ |

## D. Additional Experiments

### D.1. Computational Cost for Larger Datasets

In the case where the inner product is used as the kernel function and $n \gg d$, VendiScore can significantly reduce computational complexity. To ensure a fair comparison, we compare the computation times of VendiScore and `DCScore` on larger datasets, as well as under different kernel functions. Specifically, we employ SST2, Yelp, and AG News as the evaluated datasets. We randomly select 4k, 8k, 16k, 32k, 64k samples and record the computation times for both methods across these different sample sizes. We repeat the experiments 5 times to report the mean and standard deviation.

As shown in Tables 6, 10, and 11, `DCScore` has a shorter computation time than VendiScore in most cases, with VendiScore only exhibiting a computational advantage when the inner product is used and $n \gg d$. Furthermore, as the sample size increases, the efficiency advantage of `DCScore` becomes more pronounced. When using a polynomial kernel, on the SST2 dataset, `DCScore` requires only one-third of the computation time of VendiScore when the sample size reaches 64k. In contrast, although VendiScore has a computational advantage in the case of the inner product, the difference compared to `DCScore` is not significant. The experimental results are consistent with our complexity analysis presented in Section 4.3. Overall, `DCScore` outperforms VendiScore in terms of computation time across most kernel functions. VendiScore exhibits a computational time advantage only when the inner product is used as the kernel function, which will limit its applicability. As shown in Chapter 4 of (Seeger, 2004), it is essential to employ different kernel functions to accommodate a wider range of scenarios.

### D.2. Comparison of Computational Cost between `DCScore` and Efficiency-improved Methods

A recent work (Ospanov et al., 2024) employs the random Fourier features framework to reduce the computational cost of VendiScore (Dan Friedman & Dieng, 2023) and RKE (Jalali et al., 2023). To further investigate the computational efficiency of `DCScore`, we compare it with efficiency-improved versions of two entropy-based diversity evaluation methods, namely FKEA-Vendi and FKEA-RKE. Here, we set $\alpha$ to 1 for FKEA-Vendi and 2 for FKEA-RKE. Notably, we follow the

*Table 12.* Comparison of computation time between `DCScore` and two efficiency-improved methods (FKEA-Vendi and FKEA-RKE).

| Datasets | Sample num | 4k | 8k | 16k | 32k | 64k |
|---|---|---|---|---|---|---|
| | FKEA-Vendi | $18.43_{\pm 0.14}$ | $36.97_{\pm 0.03}$ | $74.03_{\pm 0.20}$ | $147.78_{\pm 0.30}$ | $295.14_{\pm 0.41}$ |
| **SST2** | FKEA-RKE | $18.46_{\pm 0.08}$ | $37.08_{\pm 0.12}$ | $73.98_{\pm 0.09}$ | $147.99_{\pm 0.09}$ | $297.46_{\pm 2.02}$ |
| | DCScore | $\mathbf{3.26}_{\pm 0.08}$ | $\mathbf{6.75}_{\pm 0.08}$ | $\mathbf{13.94}_{\pm 0.06}$ | $\mathbf{31.32}_{\pm 0.34}$ | $\mathbf{90.36}_{\pm 1.24}$ |
| | FKEA-Vendi | $26.28_{\pm 0.13}$ | $52.69_{\pm 0.24}$ | $106.59_{\pm 2.25}$ | $212.52_{\pm 2.22}$ | $\mathbf{422.51}_{\pm 1.19}$ |
| **Yelp** | FKEA-RKE | $\mathbf{26.19}_{\pm 0.13}$ | $\mathbf{52.68}_{\pm 0.24}$ | $\mathbf{105.27}_{\pm 0.22}$ | $\mathbf{212.38}_{\pm 1.87}$ | $422.94_{\pm 1.55}$ |
| | DCScore | $37.53_{\pm 0.08}$ | $75.19_{\pm 0.10}$ | $151.59_{\pm 0.39}$ | $306.80_{\pm 0.33}$ | $641.60_{\pm 0.60}$ |
| | FKEA-Vendi | $20.64_{\pm 0.05}$ | $41.66_{\pm 0.06}$ | $83.02_{\pm 0.11}$ | $165.92_{\pm 0.25}$ | $338.16_{\pm 14.26}$ |
| **AG News** | FKEA-RKE | $20.68_{\pm 0.04}$ | $41.68_{\pm 0.03}$ | $83.04_{\pm 0.14}$ | $165.85_{\pm 0.23}$ | $331.58_{\pm 0.29}$ |
| | DCScore | $\mathbf{10.29}_{\pm 0.49}$ | $\mathbf{21.12}_{\pm 0.73}$ | $\mathbf{43.84}_{\pm 1.02}$ | $\mathbf{91.64}_{\pm 0.86}$ | $\mathbf{213.30}_{\pm 2.15}$ |

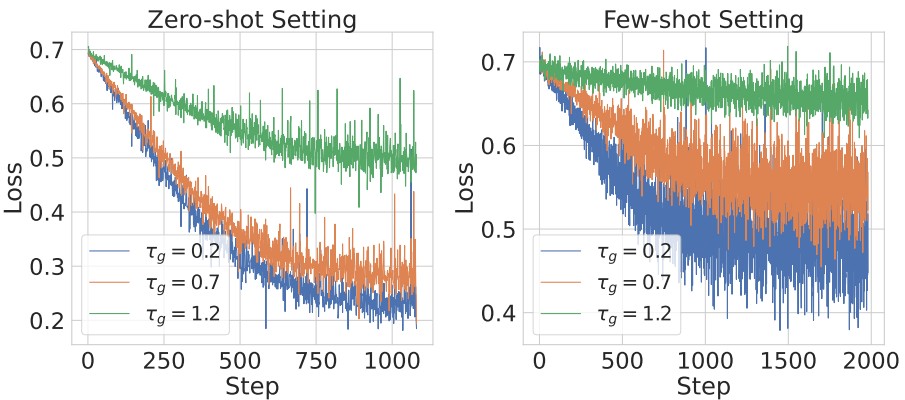

*Figure 8.* Loss curves of the downstream task training.

experimental settings described in Appendix D.1 and leverage different random seeds for data sampling. Table 12 illustrates the computation time comparison across the SST2, Yelp, and AG News datasets. `DCScore` demonstrates lower computation time compared to FKEA-Vendi and FKEA-RKE on the SST2 and AG News datasets. However, the results are reversed for the Yelp dataset. Additionally, we observe that FKEA-Vendi does not improve computation time compared to VendiScore, possibly because FKEA focuses solely on reducing the complexity of the summarization (eigenvalue computation) phase, whereas our computation time encompasses the entire calculation process.

### D.3. Correlation with Downstream Task Training

To investigate the correlation between `DCScore` and downstream task training, we train text classification models using self-generated datasets under zero-shot and few-shot settings. We vary the generation temperature $\tau_g$ of self-generated datasets within the range of $\{0.2, 0.7, 1.2\}$. More details of training datasets and hyperparameters are presented in Appendix C.1.1 and C.2.2, respectively. Figure 8 shows the loss curves of these trained classification models. In the zero-shot setting, we observe increasing optimal loss values as $\tau_g$ varied from 0.2 to 1.2, indicating that the model is more easily fitted to datasets with limited diversity. However, as shown in Table 7, models trained on more diverse datasets achieve better accuracy, which can be attributed to their enhanced generalization capabilities. From Table 7, the diversity evaluated by `DCScore` has a similar trend to the accuracy performance in the zero-shot setting, further demonstrating the effectiveness of `DCScore` in diversity evaluation.

In the few-shot setting, we observe a trend in optimal loss variation similar to that in the zero-shot setting, as shown in Figure 8. However, Figure 9 reveals that at epoch 120, models trained on datasets generated with $\tau_g = 0.7$ outperform those

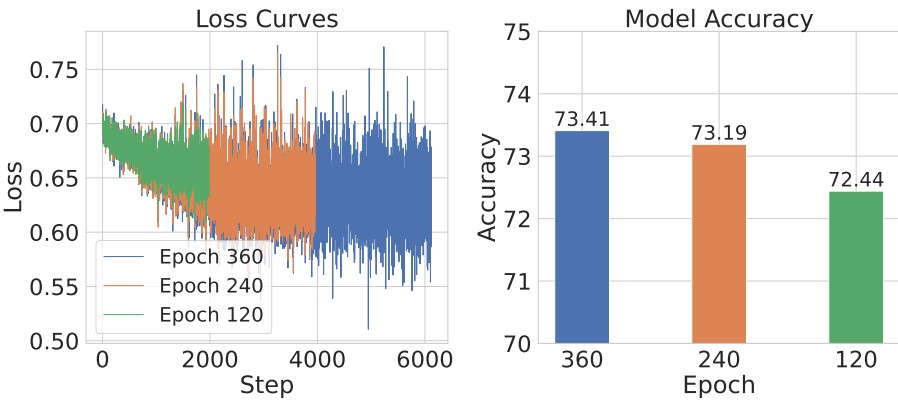

*Figure 9.* Loss curves and accuracy of models trained on generated dataset with $\tau_g = 1.2$.

*Table 13.* Correlation (Spearman's $\rho$) results of `DCScore` with various embedding functions. Spearman's $\rho$ varies between -1 and +1, with 0 implying no correlation. Best results are indicated in **bold**.

| Embedding models | Zero-shot setting | | | | Few-shot setting | | | |
|---|---|---|---|---|---|---|---|---|
| | Text classification | | Story completion | | Text classification | | Story completion | |
| | 13B | 70B | 13B | 70B | 13B | 70B | 13B | 70B |
| **SimCSE (unsup-simcse-bert-base-uncased)** | **0.9961** | 0.9779 | 0.9844 | 0.9792 | **0.9909** | 0.9883 | 0.9857 | **0.9974** |
| **SimCSE (sup-simcse-roberta-large)** | 0.9909 | 0.9753 | 0.9883 | 0.9883 | 0.9792 | **0.9935** | 0.9779 | 0.9623 |
| **Sentence BERT (all-mpnet-base-v2)** | 0.9896 | 0.9740 | 0.9870 | 0.9909 | 0.9766 | 0.9870 | 0.9857 | 0.9870 |
| **BGE (bge-large-en-v1.5)** | 0.9909 | **0.9896** | **0.9922** | **0.9948** | 0.9857 | 0.9922 | **0.9870** | 0.9922 |

using $\tau_g = 1.2$. This phenomenon can be attributed to the higher diversity of datasets generated at a higher $\tau_g$, resulting in increased fitting difficulty. Under the current settings, the number of training epochs for the dataset generated at a temperature of 1.2 is insufficient, preventing the trained model from achieving optimal performance. To validate this hypothesis, we increase the number of epochs to 240 and 360 and train models on the dataset generated at a temperature of 1.2. The final training loss and accuracy of these models are shown in Figure 9. We observe that as the number of epochs increases, the model's loss gradually decreases, and its performance improves progressively. Ultimately, the model's accuracy outperforms that of models trained on datasets generated at temperatures of 0.2 and 0.7. Moreover, from Table 7, models trained on datasets from the zero-shot setting outperform those trained on datasets from the few-shot setting. However, this discrepancy arises from the different test sets used in the two settings, making direct performance comparisons inappropriate.

### D.4. Impact of Embedding Functions Φ

The paradigm of the transformation-based method enables `DCScore` to utilize various embedding functions tailored to different scenarios. Consequently, we investigate the impact of embedding functions on self-generated datasets used in Section 5.2.1. As shown in Table 13, we compare the correlation of the diversity evaluation results of `DCScore` across 4 different embedding functions with diversity pseudo-truths, where the model names in parentheses within the embedding function refer to those available on Hugging Face. Our findings indicate that `DCScore` exhibits strong correlations with diversity pseudo-truths across various embedding functions. Notably, `DCScore` utilizing the BGE embedding function achieves the best results in half of the cases. Additionally, the minimum correlation in Table 13 exceeds 0.96, which is classified as a strong correlation according to (Akoglu, 2018). This result also supports the following two conclusions: (1) the embedding function used effectively captures the differences among samples from multiple perspectives, and (2) `DCScore` is sufficiently adaptable to different embedding functions while maintaining stable performance.

*Table 14.* Correlation (Spearman's $\rho$) results of `DCScore` with various kernel functions. Spearman's $\rho$ varies between -1 and +1, with 0 implying no correlation. Best results are indicated in **bold**.

| Embedding models | Zero-shot setting | | | | Few-shot setting | | | |
| --- | --- | --- | --- | --- | --- | --- | --- | --- |
| | Text classification | | Story completion | | Text classification | | Story completion | |
| | 13B | 70B | 13B | 70B | 13B | 70B | 13B | 70B |
| **Inner product** | **0.9961** | 0.9779 | 0.9844 | **0.9792** | **0.9909** | **0.9883** | **0.9857** | **0.9974** |
| **laplacian kernel** | 0.9935 | **0.9831** | 0.9883 | 0.9727 | 0.9597 | 0.9649 | 0.9701 | 0.9922 |
| **RBF kernel** | 0.9935 | 0.9818 | **0.9896** | 0.9753 | 0.9740 | 0.9727 | 0.9792 | 0.9922 |
| **polynomial kernel** | 0.9870 | 0.9584 | 0.9714 | 0.9506 | 0.9182 | 0.9182 | 0.9857 | 0.9896 |

### D.5. Impact of Kernel Functions

Similar to Appendix D.4, we investigate the impact of different kernel functions on the performance of `DCScore`. Specifically, this experimental setup is identical to that in Appendix D.4. As shown in Table 14, we find that `DCScore` demonstrates stable performance across various kernel functions. However, the influence of the kernel function is slightly more pronounced than that of the embedding function, as indicated by the greater fluctuations in correlation among the different kernel functions. Furthermore, we observe that `DCScore` achieves optimal performance in the case of the inner product. Overall, `DCScore` consistently maintains strong diversity evaluation performance across different kernel functions.

## E. Baseline Methods

We present the detailed modeling of baseline methods into `DCScore` as follows:

**Distinct-n.** *Distinct-n* (Li et al., 2015) is a prevalent diversity metric depending on n-grams, where *n* signifies the number of successive items. *Distinct-n* calculates the proportion of unique n-grams to all n-grams. The n-grams operation falls under the text representation stage, while the step of obtaining a unique set of n-grams corresponds to the pairwise similarity stage. Typically, a high form similarity among samples in the evaluated dataset results in a smaller unique n-gram set. Finally, ratio calculations belong to the diversity summarization stage.

$$
\begin{aligned}
&\textit{Text Representation: } \text{n-grams}(\text{Concat}(\mathcal{D})), \\
&\textit{Pairwise Similarity: } \text{Unique}(\text{n-grams}(\text{Concat}(\mathcal{D}))), \\
&\textit{Diversity Summarization: } \text{Distinct-n}(\mathcal{D}) = \frac{|\text{Unique}(\text{n-grams}(\text{Concat}(\mathcal{D})))|}{|\text{n-grams}(\text{Concat}(\mathcal{D}))|},
\end{aligned}
\tag{18}
$$

where n-grams, Unique, and Concat represent the n-grams, de-overlap process, and concatenate operation, respectively.

**K-means inertia.** K-means Inertia (Du & Black, 2019), a transformation-based method, performs clustering in sample representation and then calculates inertia as diversity outcomes. Here, inertia refers to the square summation of the distance between samples and cluster centroids.

$$
\begin{aligned}
&\textit{Text Representation: } \mathbf{H} = \Phi(\{\tilde{\mathcal{T}}_i\}_{i=1}^n), \\
&\textit{Pairwise Similarity: } \mathcal{C} = \text{K-means}(\mathbf{H}), \\
&\textit{Diversity Summarization: } \text{Inertia}(\mathcal{D}) = \sum_{\mathbf{c}_k \in \mathcal{C}, \mathbf{h}_j \in \mathbf{H}_{\mathbf{c}_k}} (\mathbf{h}_j - \mathbf{c}_k)^2,
\end{aligned}
\tag{19}
$$

where $\mathbf{H}$ is the representation of all samples and $\mathbf{h}_i$ is the representation of the $i$-th sample, $\mathcal{C}$ denotes the cluster centroid set, and $\mathbf{c}_k \in \mathcal{C}$ represents the $k$-th cluster centroid. The sample representation associated with the $k$-th cluster centroid is expressed as $\mathbf{h}_j \in \mathbf{H}_{\mathbf{c}_k}$, while $\mathbf{H}_{\mathbf{c}_k}$ denotes the sample representations corresponding to the $k$-th cluster centroid.

**VendiScore.** VendiScore (Dan Friedman & Dieng, 2023) is a recently proposed diversity metric that falls under the category of the transformation-based method. Based on sample representations, VendiScore utilizes a kernel function to calculate

a similarity matrix $\mathbf{K}$. Subsequently, VendiScore summarizes diversity as the exponential of the Shannon entropy of the eigenvalues of $\mathbf{K}/n$.

$$
\begin{aligned}
\textit{Text Representation: } & \mathbf{H} = \Phi(\{\tilde{\mathcal{T}}_i\}_{i=1}^n), \\
\textit{Pairwise Similarity: } & \mathbf{K} = \mathrm{Kernel}(\mathbf{H}), \\
\textit{Diversity Summarization: } & \mathrm{VS}(\mathcal{D}) = \exp\left(-\sum_{i=1}^n \lambda_i \log \lambda_i\right),
\end{aligned}
\tag{20}
$$

where $\mathrm{Kernel}(\cdot)$ is the kernel function, such as the inner product, $\lambda_i$ is the $i$-th eigenvalue of $\mathbf{K}/n$.

