# OpenReview forum: "Measuring Diversity in Synthetic Datasets"
_ICML.cc/2025/Conference — ICML 2025 poster_

### Official Review · Reviewer_QL32 · 2025-02-16

**Overall Recommendation:** 5

**Summary:**

The paper introduces DCScore, a novel method for measuring diversity in synthetic datasets from a classification perspective. From my analysis, the key innovation is reformulating diversity evaluation as a sample classification task, where each sample should be distinguishable enough to form its own class. The authors demonstrate this approach satisfies important theoretical properties while outperforming existing metrics across multiple datasets and evaluation criteria.

**Claims And Evidence:**

The central claims about DCScore's effectiveness are well-supported through extensive experiments across different evaluation scenarios. From my experience, the reported correlations (Spearman's ρ > 0.96) with multiple diversity pseudo-truths are quite strong. However, it would be better to add more investigation of failure cases and limitations

**Essential References Not Discussed:**

The paper should discuss recent work on classification-based evaluation metrics, particularly "Beyond Accuracy: Behavioral Testing of NLP Models with CheckList" (Ribeiro et al., 2020) which provides relevant insights about classification-based evaluation.

**Experimental Designs Or Analyses:**

From my perspective, a key strength is using multiple correlation measures (τg, human, LLM) as pseudo-truths. However, it would be better to add diversity evaluations on more real-world synthetic datasets beyond those augmented by LLMs.

**Methods And Evaluation Criteria:**

From my analysis, the validation of theoretical properties (effective number, symmetry, etc.) strongly supports the method's soundness. However, it would be better to add more discussion of the hyperparameter sensitivity of the classification temperature τ.

**Other Comments Or Suggestions:**

NA

**Other Strengths And Weaknesses:**

The key strength is the novel classification perspective that provides both theoretical guarantees and strong empirical results. The main weakness is limited discussion of the method's applicability to non-text modalities.

**Questions For Authors:**

How does the method perform on multi-modal synthetic data where text is just one component?
What is the sensitivity of the method to the choice of classifier architecture beyond the examined options?

**Relation To Broader Scientific Literature:**

The work builds meaningfully on existing diversity metrics while making novel contributions. From my experience, it would be better to add discussion of connections to other classification-based metrics in machine learning beyond just diversity measurement.

**Theoretical Claims:**

I have carefully reviewed the theoretical foundations and proofs in Section 4.2 and Appendix B. The axiomatic guarantees (effective number, identical samples, symmetry, monotonicity) are mathematically sound and well-proven. The complexity analysis comparing with VendiScore is thorough.

---

> ### Author Rebuttal · Authors · 2025-04-01
>
> We appreciate the reviewer's recognition of our work. We respond to the reviewer’s question as follows. **Limited by the space, we present our additional experiments in an anonymous URL** (https://anonymous.4open.science/r/ICMLRebuttal_DCScore).
>
> ---
> >Q1: It would be better to add more investigation of failure cases and limitations
>
> R1: Thank you for your suggestive comment. **One of the primary limitations of DCScore is its inapplicability to multimodal data**. This limitation arises from the challenges associated with feature extraction and alignment across different modalities, which can affect the calculation of the classification probability matrix in DCScore. **We will include a detailed discussion of this limitation in future versions of our paper and explore potential solutions in our future research.**
>
> >Q2: It would be better to add more discussion of the hyperparameter sensitivity of the classification temperature τ.
>
> R2: We would like to clarify that we **have conducted sensitivity experiments on the classification temperature $\tau$ in Section 5.4** of our paper. These experiments explore how different values of $\tau$ influence the classification resolution. Specifically, a lower $\tau$ enhances the discriminative ability of DCScore to different samples.
>
> >Q3: It would be better to add diversity evaluations on more real-world synthetic datasets beyond those augmented by LLMs.
>
> R3: Thank you for your suggestion. We have conducted several experiments across different existing text and image datasets. Specifically, in **Figure 4 of the anonymous URL**, we present diversity scores for the AGNews/SST/Yelp_A.P. datasets augmented by AttrPrompt[1]. In **Figure 2 of the anonymous URL**, we show results for image data. Overall, DCScore shows strong correlation with baseline methods.
>
> Reference:
> [1] Large language model as attributed training data generator: A tale of diversity and bias, Neurips 2023.
>
> >Q4: From my experience, it would be better to add discussion of connections to other classification-based metrics in machine learning beyond just diversity measurement.
>
> R4: Thank you for your thoughtful review comment. We will include a discussion on classification-based metrics in machine learning in future versions of our paper. To the best of our knowledge, we **have identified classification-based methods primarily within the domain of metric learning**. To further enhance our work, we would greatly appreciate any additional guidance or references within this topic.
>
> >Q5: The paper should discuss recent work on classification-based evaluation metrics, particularly "Beyond Accuracy: Behavioral Testing of NLP Models with CheckList" (Ribeiro et al., 2020) which provides relevant insights about classification-based evaluation.
>
> R5: Thank you for your suggestion. This paper (Beyond Accuracy: Behavioral Testing of NLP Models with CheckList) primarily focuses on guiding users in conducting Behavioral Testing of NLP Models, but it appears to not mention classification-based evaluation metrics. However, we recognize the relevance of this work to our paper and **will include a discussion of its insights in the related work section of the future version.**
>
> >Q6:  The main weakness is limited discussion of the method's applicability to non-text modalities.
>
> R6: Thank you for your valuable review. We conduct experiments on the image modality (colored mnist dataset), please refer to **Figure 2 of the anonymous URL**. We follow the setting of [1] and observe that **DCScore presents higher correlation with the label number compared to VendiScore.**
>
> Reference:
> [1] Ospanov et al., Towards a Scalable Reference-Free Evaluation of Generative Models, NeurIPS 2024.
>
> >Q7: 1.How does the method perform on multi-modal synthetic data where text is just one component? 2.What is the sensitivity of the method to the choice of classifier architecture beyond the examined options?
>
> R7: For 1, our method has limitations in evaluating multi-modal synthetic data, which is an area we plan to explore further. Specifically, **the effectiveness of DCScore in accurately evaluating multi-modal data depends on the extraction of multi-modal representations and the alignment of different data modalities**. We believe this is a very worthwhile research direction and appreciate your guidance on this matter.
>
> For 2, in addition to the factors we have already explored, we believe **the sensitivity of the classifier architecture lies in its ability to effectively distinguish differences between samples**. The fundamental requirement for our method to function correctly is that the classifier can accurately identify distinct samples.

---

### Official Review · Reviewer_fn8g · 2025-02-28

**Overall Recommendation:** 3

**Summary:**

- The paper introduces DCScore, a novel method for measuring diversity in synthetic datasets generated by large language models (LLMs).

- Key innovation: DCScore formulates diversity evaluation as a sample classification task, leveraging mutual relationships among samples, rather than using traditional n-gram statistics or reference-based methods.

- Main contribution: A principled diversity evaluation metric that satisfies important diversity-related axioms (effective number, identical samples, symmetry, and monotonicity).

- Technical approach: The method maps diversity-sensitive components into a representation space, computes pairwise similarities through a kernel function, and summarizes diversity through classification probabilities.

- Experimental validation: DCScore demonstrates stronger correlation with diversity pseudo-truths (like generation temperature) and human judgment compared to baseline methods.

- Computational efficiency: Both theoretical analysis and empirical results show DCScore has lower computational costs compared to existing approaches, particularly for non-linear kernels.

**Claims And Evidence:**

- The theoretical claims about DCScore satisfying the four axioms (effective number, identical samples, symmetry, monotonicity) are well-supported with formal proofs in the paper.

- The computational efficiency claims are supported by both theoretical complexity analysis (Table 2) and empirical measurements (Figure 4), though the advantage varies by kernel type.

- The claim about DCScore's correlation with human judgment is supported (Table 4), but would be strengthened with more details on the human evaluation protocol and inter-annotator agreement.

- The claim about correlation between DCScore and downstream task performance is supported by limited evidence (Table 7) that could benefit from more extensive experimentation. Additionally, it seems like the number of epochs was not fixed for this experiment.

**Essential References Not Discussed:**

N/A

**Experimental Designs Or Analyses:**

- The correlation analysis with generation temperature (τg) is sound, using appropriate statistical measures (Spearman's ρ) across a well-distributed range of temperatures (0.2-1.2).

- The experimental setup for human evaluation lacks some important details - specifically how agreement was measured across evaluators and why only 3 annotators were used.

- The downstream task training experiment provides valuable real-world validation, but could be strengthened with more diverse task types beyond text classification.

- The batch evaluation protocol (averaging results across batches generated from the same context) is appropriate but could introduce bias if contexts vary significantly in diversity potential.

- The ablation studies for embedding functions and kernel types are methodologically sound, though the paper could better explain the criteria for selecting the specific functions tested.

- The comparison against baseline methods is fair, with appropriate implementation details provided for each method.

- The paper appropriately tests on both self-generated and publicly available datasets, but the dataset sizes (100 samples per sub-dataset in some experiments) are somewhat limited for diversity evaluation.

- Statistical significance testing is notably absent from the correlation analysis, which would strengthen confidence in the reported differences between methods.

**Methods And Evaluation Criteria:**

- Using Spearman's ρ to measure correlation with diversity pseudo-truths is appropriate for evaluation.

- The selection of diversity pseudo-truths (generation temperature, human judgment, GPT-4 evaluation) is sensible and provides multiple validation angles.

- The choice of baseline comparison methods (Distinct-n, K-means inertia, VendiScore) covers multiple baseline approaches.

- The evaluation on both self-generated and publicly available datasets is appropriate, though more domain diversity would strengthen claims of generalizability.

- Testing across multiple embedding functions and kernel types demonstrates robustness, a valuable evaluation approach.

- The computational efficiency evaluation is practical and relevant, especially for large synthetic datasets.

- The axiomatic analysis provides theoretical validation for why the method works, strengthening the methodology.

- A comparison with reference-free diversity metrics from other domains (e.g., ecology) could have further contextualized DCScore's advantages.

**Other Comments Or Suggestions:**

- Figure 4 is very small and is quite hard to read.

**Other Strengths And Weaknesses:**

## Strengths

- The paper addresses a practical need in LLM-generated datasets, making it timely and relevant to current research directions.

- The method is adaptable across embedding functions and kernel types, demonstrating flexibility for different applications.

- The computational efficiency advantages make DCScore practically applicable to large-scale dataset evaluation.

## Weaknesses

- Limited experimental evaluation across diverse domains beyond text classification and story completion.

- Human evaluation is sparse.

- Limited discussion of potential failure cases or situations where DCScore might not accurately capture diversity.

- Computational results focus on small to medium datasets (up to 64k); scalability for extremely large datasets remains unproven.

**Questions For Authors:**

1. Can you provide more details about the human evaluation protocol? Specifically, how was inter-annotator agreement measured, and what were the specific instructions given to evaluators?

2. Have you tested DCScore on non-basic-text modalities (e.g., code, mathematical expressions, structured data)? If not, what adaptations would be necessary to apply it to these domains?

3. How does DCScore perform on datasets with substantial outliers or on highly imbalanced datasets where certain types of content dominate?

4. Have you explored whether DCScore can be used for diversity-aware sampling from generative models, beyond just evaluation of already-generated datasets?

**Relation To Broader Scientific Literature:**

- The paper connects to recent LLM synthetic-data generation work by Ye et al. (2022) and Abdullin et al. (2024), providing evaluation metrics for these generative approaches.

- The computational efficiency focus addresses challenges with other methods (such as VendiScore).

- The evaluation methodology connects to research on human-aligned evaluation metrics by Holtzman et al. (2019).

- The temperature-based diversity relation builds on sampling strategy research by Caccia et al. (2018) and Chung et al. (2023).

**Theoretical Claims:**

I did not check the correctness of proofs very carefully. Overall, they seem mostly correct. They could better explore potential limitations in edge cases, such as what happens with extremely imbalanced datasets or outliers.

---

> ### Author Rebuttal · Authors · 2025-04-01
>
> We thank the reviewer for reading our paper. We respond to the reviewer’s question as follows. **Limited by the space, we present our additional experiments in an anonymous URL** (https://anonymous.4open.science/r/ICMLRebuttal_DCScore).
>
> ---
> >Q1: The claim about DCScore's correlation with human judgment would be strengthened with more details on the human evaluation protocol and inter-annotator agreement.
>
> R1: Thank you for your valuable comment. **We will update the appendix in the subsequent version of our paper to include details of the human evaluation protocol**. Human evaluation data were generated at six temperatures, with five results per context (prompt) in each subset. During the evaluation, annotators were asked to select the more diverse subset from pairs of subsets. Across six temperatures, this resulted in 15 comparisons, and with three annotators, a total of 45 judgments were made. Subsets were ranked by the frequency of being chosen as more diverse. This process was repeated five times with different contexts to derive the final human diversity ranking.
>
> >Q2: The claim about correlation between DCScore and downstream task performance could benefit from more extensive experimentation. Additionally, it seems like the number of epochs was not fixed for this experiment.
>
> R2: We** have included some downstream task experiments in Appendix E.2. In Table 7**, except for the scenario with $\tau_{g}=1.2$(360 epochs), all other results were obtained with 120 epochs. **The higher dataset diversity, e.g. datasets with $\tau_{g}=1.2$, required more epochs for model convergence**, with details in Appendix E.2. **Figure 8 also presents results for $\tau_{g}=1.2$ at 240 and 120 epochs**.
>
> >Q3: A comparison with reference-free diversity metrics from other domains (e.g., ecology) could have further contextualized DCScore's advantages.
>
> R3: Our method is primarily concerned with evaluating the diversity of synthetic datasets. We believe that **comparisons with methods from more closely related domains might carry more conviction**. The axiomatic requirements for diversity evaluation can vary significantly across different fields, and using metrics from ecology to assess text or image datasets might result in an unfair comparison. Additionally, **the main idea behind the Vendi score method originates from ecology, which can be considered an indirect comparison with metrics from other fields.**
>
> >Q4:  Limited discussion of potential failure cases or situations where DCScore might not accurately capture diversity.
>
> R4: The main limitation of our method is that **it is not applicable to multimodal data**. Due to space constraints, please refer to **R1 of the Response to Reviewer QL32** for more details.
>
> >Q5: The batch evaluation protocol is appropriate but could introduce bias if contexts vary significantly in diversity potential.
>
> R5: The batch evaluation protocol was chosen to ensure that the diversity pseudo-truths are accurately reflected when generating multiple samples from the same context. This approach helps maintain consistency in evaluating the diversity of generated datasets. We adopted this evaluation strategy to conduct correlation experiments, and under this setting, bias is not introduced.
>
> >Q6: The paper appropriately tests on both self-generated and publicly available datasets, but the dataset sizes (100 samples per sub-dataset) are somewhat limited for diversity evaluation.
>
> R6: As shown in Q8, we conducted experiments on datasets with **sizes reaching up to 64k samples.** The smaller datasets, consisting of 100 samples per sub-dataset, were specifically designed for batch evaluation protocols (**Please refer to R5** for more details).
>
> >Q7: Limited experimental evaluation across diverse domains beyond text classification and story completion.
>
> R7: Thank you for your suggestion. We provide experimental results on image data (colored mnist dataset), please refer to **Figure 2 of anonymous URL**. DCScore presents a higher correlation with the label number compared to VendiScore.
>
> >Q8: Computational results focus on small to medium datasets (up to 64k); scalability for extremely large datasets remains unproven.
>
> R8: We provide experimental results on extremely large datasets (up to 120k) in **Figure 6 of the anonymous URL. Notably, DCScore exhibits similar changing trends with the baseline methods across these large-scale datasets**.
>
> >Q9: Have you explored whether DCScore can be used for diversity-aware sampling from generative models, beyond just evaluation of already-generated datasets?
>
> R9: Our method is similar to approaches like the Vendi score, thus **it can be applied to diversity-aware sampling from generative models**. However, we are more focused on scenarios of synthetic data diversity evaluation. We also provide a detailed discussion of potential application scenarios for DCScore in Appendix A.2.

---

### Official Review · Reviewer_BZYd · 2025-03-13

**Overall Recommendation:** 2

**Summary:**

The paper introduces DCScore, a novel metric for measuring diversity in synthetic datasets. Unlike traditional methods (e.g., Distinct-n, VendiScore), DCScore models diversity as a classification task and uses semantic embeddings to compute pairwise similarity among samples. It leverages a softmax-based classification probability matrix to quantify dataset diversity. The authors provide theoretical guarantees for DCScore, showing that it satisfies fundamental diversity axioms. Empirical results demonstrate that DCScore correlates strongly with human judgments while being more computationally efficient than VendiScore.

**Claims And Evidence:**

This paper assumes that classification probability correlates directly with diversity, but it does not sufficiently justify why treating each sample as a separate category is a robust measure of diversity.If classification probability is used as a proxy for diversity, the paper should provide empirical or theoretical validation, such as, showing that classification-based diversity correlates well with entropy-based or clustering-based diversity measures.
The paper correctly highlights that VendiScore requires O(n3) complexity due to eigenvalue decomposition, while DCScore reduces this to O(n2).However, VendiScore can be optimized to O(d 2n)using low-rank approximations, which the authors do not address.

**Essential References Not Discussed:**

The paper does not discuss entropy-based diversity metrics, which have been used in representation learning and active learning.

**Experimental Designs Or Analyses:**

There is no ablation study on how different embedding functions affect DCScore, the author should analyze how different embeddings (e.g., SBERT, CLIP) affect DCScore.

**Methods And Evaluation Criteria:**

The method of treating diversity evaluation as a classification problem is interesting, but it is not fully justified.
The method assumes that classification probability correctly represents diversity, but this assumption is not tested against other diversity definitions (e.g., entropy-based or clustering-based measures).
No test on datasets with strong semantic similarity (e.g., paraphrased text or redundant images), which would be important to validate the method.

**Other Comments Or Suggestions:**

Please see the above weakness

**Other Strengths And Weaknesses:**

Strengths
1.Proposes a novel classification-based diversity metric.
2.Improves computational efficiency over VendiScore.
Weaknesses:
1.Lacks theoretical justification for using classification probability.
2.Highly sensitive to how categories are defined.
3.Does not compare with entropy-based or contrastive learning-based diversity metrics.

**Questions For Authors:**

Please see the above weakness

**Relation To Broader Scientific Literature:**

The paper discusses N-gram-based, reference-based, and transformation-based diversity metrics.However, it does not mention entropy-based diversity measures, which are commonly used in active learning and generative models.

**Theoretical Claims:**

The paper propose DCScore，but if classification probability does not fully represent diversity, then the theoretical guarantees may be incomplete.

---

> ### Author Rebuttal · Authors · 2025-04-01
>
> We thank the reviewer for reading our paper. We respond to the reviewer’s question as follows. **Limited by the space, we present our additional experiments in an anonymous URL** (https://anonymous.4open.science/r/ICMLRebuttal_DCScore).
>
> ---
> >Q1: it does not sufficiently justify why treating each sample as a separate category is a robust measure of diversity. the paper should provide that classification-based diversity correlates well with entropy-based or clustering-based diversity measures.
>
> R1: Thank you for your insightful comments. **We have indeed provided both experimental and theoretical validation to support DCScore as a robust measure of diversity.** We offer empirical validation demonstrating a strong correlation with the entropy-based method, as shown in **Figure 5 at the anonymous URL**.
>
> Theoretically, we have proven that DCScore satisfies several axioms that an ideal diversity evaluation method should meet, as detailed in Section 4.2 and Appendix B of our paper.
>
> Empirically, we have demonstrated that DCScore has a high correlation with multiple diversity pseudo-truths, such as generation temperature $\tau_{g}$, human evaluation, and LLM evaluation, as shown in Section 5.2. We believe these pseudo-truths more accurately reflect the true diversity of datasets.
>
> >Q2: VendiScore can be optimized to O(d 2n)using low-rank approximations, which the authors do not address.
>
> R2: As shown in Table 2 of our paper, **VendiScore can achieve $\mathcal{O}(d^{2}n)$ complexity when using a linear kernel**, such as Inner Product. We have provided a detailed analysis of this in Section 4.3. However, in practical evaluation scenarios, the diversity of synthetic datasets often requires more complex kernels beyond linear ones.
>
> >Q3:  1.Classification probability correctly represents diversity, but this assumption is not tested against other diversity definitions (e.g., entropy-based measures). 2.No test on datasets with strong semantic similarity.
>
> R3: For 1, DCScore and entropy-based or clustering-based measures operate on fundamentally different principles, **making it infeasible to validate one method's assumptions within the framework of another**. Limited by the space, please refer to R4 for more details.
> For 2, as shown in **Figure 1 of anonymous URL**, we conduct evaluation experiments on datasets with strong semantic similarity and **observe that DCScore performs well in this scenario**.
>
> Reference:
> [1] Ospanov et al., Towards a Scalable Reference-Free Evaluation of Generative Models, NeurIPS 2024.
>
> >Q4: If classification probability does not fully represent diversity, then the theoretical guarantees may be incomplete. The author should add mathematical proofs for why classification probability is a good diversity metric
>
> R4: Thank you for your suggestion. For diversity metrics, **there is no ground truth or golden rule, only characterization through axioms**. Therefore, in Section 4.2, we demonstrate that DCScore satisfies these axioms, indicating that it is a good diversity evaluation metric. From the theoretical perspective, Diversity evaluation aims to capture dataset richness by identifying sample relationships, which aligns with the classification perspective of DCScore. Thus, **a sample-level classification probability can fully represent diversity**.
>
> >Q5: There is no ablation study on how different embedding functions affect DCScore, the author should analyze how different embeddings (e.g., SBERT, CLIP) affect DCScore
>
> R5: We would like to clarify that **we have conducted experiments on the impact of embedding functions**, and the results are presented in **Appendix E.3 and Table 11**. DCScore demonstrates strong performance across various embedding functions.
>
> >Q6: The supplementary material includes additional experimental details but lacks deeper theoretical analysis.
>
> R6: Thank you for your valuable feedback. In **Appendix B** of our paper, we provide a theoretical analysis of the properties satisfied by DCScore, demonstrating that it adheres to the axioms expected of an ideal diversity evaluation method. Additionally, in **Section 4.3**, we offer a theoretical analysis of the computational complexity of DCScore.
>
> >Q7: The paper does not mention entropy-based diversity measures, which are commonly used in active learning and generative models.
>
> R7: Thank you for your suggestion. We believe **the omission may stem from our different taxonomy of existing methods**. According to our classification, entropy-based diversity measures fall under Transformation-based Methods. We will clarify this distinction in future versions of our paper and include a more detailed discussion of entropy-based methods.
>
> >Q8: Weaknesses: Highly sensitive to how categories are defined
>
> R8: We would like to clarify that **DCScore is not sensitive to category definitions**. It doesn't require predefined categories and the category count is the dataset sample size.

---

### Official Review · Reviewer_xZQX · 2025-03-14

**Overall Recommendation:** 2

**Summary:**

The paper introduces a classification-based evaluation metric, DCScore, for assessing the diversity of synthetic datasets. The authors address computational challenges while satisfying axiomatic requirements and providing a holistic analysis. Additionally, they evaluate the diversity of generated datasets across various scenarios.

---

## update after rebuttal

I thank the authors for their response. I believe the authors provided more detailed insights into the advantages of their work during the rebuttal, such as the optimization stability or computational efficiency of DCScore compared to Vendi Score in some cases.

While I appreciate the authors' response, I remain concerned that several claims lack sufficient explanation or need additional experiments, hence, I will keep my score. Below are key points that, in my view, must be addressed to strengthen the manuscript:

- **Convergence of DCScore (R9)**: One important part that is missing is the convergence analysis of DCScore. The authors indeed provided additional experiments with 2D Gaussian mixtures, such analysis must also be demonstrated on complex datasets to validate the method's convergence. Also, the authors mentioned their task is distinct from generative model evaluations because the target scenario lacks ground truth values, but I believe convergence is important in their task too and even in generative models' evaluation, we don't have ground truth values.

- **Evidence of why DCScore is closer to the essence of diversity evaluation. (R4)**: While the classification-based approach is novel, the authors must provide a more rigorous comparison and evidence demonstrating why DCScore fundamentally captures the essence of diversity better than clustering-based methods like Vendi Score.

- **Fair comparison between Vendi Score with DCScore (R1, R5, R8)**: The authors claimed that DCScore is *Suitable for highly diverse scenarios*, or *In datasets with distinctive samples, Vendi Score fails, but DCScore succeeds*. However, I argue these limitations could be addressed in Vendi Score through proper hyperparameter selection.  Since DCScore has a temperature hyperparameter ($\tau$), the authors should comparably test Vendi Score using a Gaussian kernel (not cosine similarity) since Gaussian kernel has bandwidth parameter ($\sigma$) that could make it equally adaptable to such scenarios.

**Claims And Evidence:**

The paper proposes DCScore, a classification-based diversity metric for synthetic datasets. I am not convinced that using a classification method can provide new insights or even achieve results comparable to eigendecomposition methods in all cases. Using the entropy of eigenvalues can reveal the number of modes in the dataset and their underlying relationships, whereas relying solely on the trace of P may obscure important structural details.

For example, consider the Monotonicity axiom in a scenario where a large percentage of the dataset consists of duplicate samples.  The kernel matrix develops a block structure with just a few dominant eigenvalues. Using an entropy based metric on eigenvalues could capture the low diversity, whereas a trace-based approach might mistakenly indicate high diversity. This is just a simple example, however, I expect a more in-depth analysis between Vendi Score and DCScore.

**Essential References Not Discussed:**

There are several related works on reference-based methods that seem to be missing such as [1, 2]. The paper notes that DCScore is similar to the Vendi Score and compares their computational complexities in Section 4.3. While DCScore reduces the complexity from $\mathcal{O}(n^3)$ to $\mathcal{O}(n^2d + n^2)$ (except for the cosine similarity kernel), there are additional works addressing this challenge that are not cited. For instance, in [3] (see Corollary 2), it is shown that the RKE (Vendi$_{\alpha=2}$) score can be computed in $\mathcal{O}(n^2)$ for all kernels. Additionally, [4] presents an approach that estimates the Vendi score with a computational complexity of $\mathcal{O}(n)$. I believe these papers are highly correlated and should be mentioned when claiming to improve the computational complexity.

I also believe that the authors need to mention the similarity of Equation 4 (the definition of the classification probability matrix $P$), as it is similar to the contrastive learning literature. For example, in Equation 1 of the SimCLR framework [5], the authors used the same function for contrastive learning of visual representations.

[1] Naeem et al., Reliable Fidelity and Diversity Metrics for Generative Models, ICML 2020.

[2] Kynkäänniemi et al., Improved Precision and Recall Metric for Assessing Generative Models, NeurIPS 2019.

[3] Jalali et al., An Information-Theoretic Evaluation of Generative Models in Learning Multi-modal Distributions, NeurIPS 2023

[4] Ospanov et al., Towards a Scalable Reference-Free Evaluation of Generative Models, NeurIPS 2024

[5] Chen et al., A Simple Framework for Contrastive Learning of Visual Representations, ICML 2020

**Experimental Designs Or Analyses:**

The problem formulation indeed makes sense and the authors extensively experimented with different scenarios to evaluate the diversity of the synthetic dataset.

**Methods And Evaluation Criteria:**

The problem formulation of evaluating synthetic datasets indeed makes sense and the authors extensively experimented with different scenarios to evaluate the score.

**Other Comments Or Suggestions:**

I believe the authors should revise the title of each paper, as they currently use the template title “Submission and Formatting Instructions for ICML 2025.”

**Other Strengths And Weaknesses:**

**Strengths:**

- Introduce new settings for synthetic data evaluation using LLMs.

- This paper uses a classification perspective to evaluate the diversity, which is quite novel.

**Weaknesses:**

- The classification-based approach may primarily offer improved sample complexity compared to Vendi Score, without clearly demonstrating how it provides new insights into synthetic data diversity. The paper should clarify whether the classification perspective yields benefits beyond computational efficiency.

- It is not well established when or why DCScore outperforms Vendi Score. The paper needs to illustrate scenarios where Vendi Score fails to capture diversity correctly, but DCScore succeeds (Beyond computational gains)

- Although the paper emphasizes the reduction in computational complexity, it lacks a comparison with recent works that also offer efficient computations of Vendi Score. Without this, the claimed advantages of DCScore remain somewhat unsubstantiated.

**Questions For Authors:**

I believe the idea of evaluating the diversity of synthetic data is crucial, however, I am concerned about the novelty and the contribution of the paper. I have the following questions from the authors. Answering these can elaborate the contribution of this paper.

1. DCScore proposes a synthetic diversity metric with a classification perspective. Could the authors elaborate on what new insight this perspective can bring? Is it only improving the sample complexity compared to Vendi Score, or using this perspective will better match the task of diversity evaluation of synthetic datasets?

2. Using entropy of eigenvalues as a diversity measure makes sense to me as it shows how many significant directions of variation exist in the dataset, but only using the trace of the probability matrix is not very intuitive as it highly relies on individual datapoints. Could authors provide a more detailed explanation of why such a choice is a suitable metric of diversity? (One case of failure I imagine is when we add replicas already existing samples in the dataset. Could authors compare the effect of replicas on Vendi Score and DCScore?)

3. What is the advantage of DCScore over Vendi Score? Can you provide a case when DCScore behaves as we expect and Vendi score fails to do that?

4. Can the authors provide a computational complexity comparison between DCScore and the mentioned related works [3, 4] as they also provide an efficient computation of Vendi Score? Are there some cases where DCScore outperforms compared to the mentioned papers that I'm missing?


[3] Jalali et al., An Information-Theoretic Evaluation of Generative Models in Learning Multi-modal Distributions, NeurIPS 2023

[4] Ospanov et al., Towards a Scalable Reference-Free Evaluation of Generative Models, NeurIPS 2024

**Relation To Broader Scientific Literature:**

This paper uses a classification perspective to evaluate the diversity, which is quite new in the diversity evaluation literature. They also study the diversity evaluation of synthetic datasets.

**Theoretical Claims:**

Proofs of Properties of DCScore are correct and straightforward.

---

> ### Author Rebuttal · Authors · 2025-04-01
>
> We thank the reviewers for providing detailed review on our submission. We respond to the reviewers’ concerns and questions one by one. **Limited by the space, we present our additional experiments in an anonymous URL** (https://anonymous.4open.science/r/ICMLRebuttal_DCScore).
>
> ---
> >Q1: Relying solely on the trace of P may obscure important structural details. I expect a more in-depth analysis between Vendi Score and DCScore.
>
> R1: A more in-depth analysis is as follows:
> - **Computational Efficiency**: As shown in Table 2, with a general kernel, DCScore has a complexity of $\mathcal{O}(n^2)$ during summarization, while VendiScore has $\mathcal{O}(n^3)$, making DCScore more efficient.
> - **Optimization Stability**: As an optimization target, the trace of **P** (DCScore) provides simpler gradients, enabling more stable optimization. In contrast, Vendi Score can have gradient issues due to identical eigenvalues.
> - **Suitable for highly diverse scenarios**: DCScore highlights distinct samples (Please refer to R5), while Vendi Score may underestimate diversity. Thus, our approach aligns with the trend of generating diverse data.
>
> **For more advantages of our method, please refer to R4.**
>
> >Q2: There are several related works on reference-based methods that seem to be missing such as [1, 2]. There are additional works [3, 4] addressing complexity challenges that are not cited.
>
> R2: Thank you for your suggestion. **We will include the discussion of the related works you pointed out in the revised manuscript.** For paper 3, the complexity of  $\mathcal{O}(n^2)$ only considers the calculation of the Frobenius norm and excludes the specific kernel calculation due to varying complexities across different kernels (e.g. RBF kernel:$\mathcal{O}(n^2d)$, Graph Kernels (random walk):$\mathcal{O}(n^3)$).
>
> >Q3: The authors need to mention the similarity of Equation 4 , as it is similar to the contrastive learning literature, such as Equation 1 of the SimCLR.
>
> R3: Thank you for your suggestion. The core concept of contrastive learning—distinguishing samples—is similar to DCScore's classification perspective, reflecting a common idea across fields. We will highlight this in the revised manuscript.
>
> >Q4:  The paper should clarify whether the classification perspective yields benefits beyond computational efficiency. Could the authors elaborate on what new insight this perspective can bring?
>
> R4: As noted in Papers 3 and 4, **entropy-based metrics like Vendi Score face high computational costs, underscoring the need for efficiency**. Additionally, DCScore offers some new insights:
> - A novel perspective closer to the essence of diversity evaluation (identifying sample differences).
> - More stable optimization due to its simpler gradient, unlike Vendi Score's instability from eigenvalue calculations [1].
> - A clearer interpretation of sample uniqueness compared to entropy-based metrics.
> - Sensitivity to unique samples via Softmax and Trace operations, aiding outlier detection and reducing model impact.
>
> Reference:[1] Eigenvalue optimization[J]. Acta numerica, 1996.
>
> >Q5: The paper needs to illustrate scenarios where Vendi Score fails to capture diversity correctly, but DCScore succeeds.
>
> R5: Thank you for your suggestion. **In datasets with distinctive samples, Vendi Score fails, but DCScore succeeds**. In this regard, eigenvalue distributions of the kernel matrix can be highly uneven (for example, some eigenvalues are much larger than others), leading to a significant reduction in Shannon entropy. Consequently, the Vendi score yields a diversity estimate much smaller than the actual sample size. In contrast, DCScore accurately estimates the diversity of a dataset based on classification probabilities.
>
> >Q6:  1. Could authors provide a more detailed explanation of why such a choice is a suitable metric of diversity? 2. Could authors compare the effect of replicas on Vendi Score and DCScore?
>
> R6: For 1, as noted in **R1, R4, and R5**, DCScore is more computationally efficient and serves as a more stable optimization target. As theoretically shown in Section 4.2, DCScore satisfies the axioms required for an ideal diversity evaluation method.
> For 2, **DCScore does not fail in the presence of repeated samples**. According to Section 4.2 (Effective Number), we provide an proof regarding the evaluation under repeated samples. Furthermore, we provide an evaluation of repeated samples in **Figure 1 of anonymous URL**.
>
> >Q7: Can the authors provide a computational complexity comparison between DCScore and the mentioned related works [3, 4] as they also provide an efficient computation of Vendi Score?
>
> R7: We present a comparison in **Figure 3 at the anonymous URL. DCScore outperforms efficient methods in terms of computation time on the AGNews and SST datasets**. Notably, the efficient computation method from Paper 4 is incompatible with non-shift-invariant kernels, which don’t meet diversity evaluation requirements.

---

> > ### Comment · Reviewer_xZQX · 2025-04-03
> >
> > I thank the authors for their hard work and for providing additional numerical results in a short time. I have carefully read the authors' response and believe it provided additional insights into their work.
> >
> > **R1: In-depth analysis between Vendi Score and DCScore**
> >
> > - I completely agree that the paper's method is more stable for optimization compared to Vendi Score (in case where $\alpha \ne 2$). When $\alpha=2$ (RKE), it can be computed using Frobenius norm and it is stable.
> >
> > - Suitable for highly diverse scenarios: May I ask for more clarification on why the authors claim that the Vendi score failed in the replica experiment because the "diversity estimate was much smaller than the actual sample size"? Replicating exact samples does not add diversity (just imagine a model that outputs replication of the same image), and the Vendi score captured this scenario. Also, choosing the kernel bandwidth plays a significant role here (just like the temperature hyperparameter in DCScore).
> >
> > **R4: New insights of DCScore**
> >
> > I thank the authors for providing these insights. It will definitely improve the draft by supporting these numerically.
> >
> > **R5: Scenarios where CDScore has more advantages**:
> >
> > As I mentioned in R1, I did not quite understand why Vendi failed and DCScore succeeded. Also, this raises a concern regarding the **sample convergence** of DCScore. How many samples do we need for DCScore to converge? (I do not expect numerical results for this concern at this point and just want to emphasize the importance of addressing this point.)
> >
> > **R7:** Could the authors clarify what settings they used (e.g., how many RFFs they used)? I appreciate the point about the summerization part complexity, but isn't the bottleneck for the Vendi score?
> >
> > **R7: non-shift-invariant kernels don’t meet diversity evaluation requirements**
> >
> > I would like to show my concern regarding this statement. I believe shift-invariant kernels are necessary for diversity evaluation.
> > When the kernel is not shift-invariant, the diversity score changes when you shift the data, even though the actual diversity of the data doesn't change. That's why shift-invariant kernels are necessary for proper diversity evaluation. Could the authors elaborate on why this fails to meet diversity evaluation requirements?

---

> > > ### Author Response · Authors · 2025-04-08
> > >
> > > Thank you for your comments. We respond to the reviewer’s question as follows. **Due to differences in the experimental setup, we spent some time conducting sample convergence experiments**. We would appreciate it if you could take our responses into consideration when making the final evaluation of our work.
> > >
> > > ---
> > > >Q8: May I ask for more clarification on why the authors claim that the Vendi score failed in the replica experiment because the "diversity estimate was much smaller than the actual sample size"?
> > >
> > > R8: There seems to be a misunderstanding. **We do not claim that the Vendi Score fails in the replica experiment.** Perhaps the reviewer intended to ask **why the Vendi Score fails in highly diverse scenarios**. We provide a detailed explanation below:
> > >
> > > - As indicated in R5, highly diverse scenarios involve more distinctive samples (dissimilar samples), which can lead to uneven eigenvalue distributions (where some eigenvalues are much larger than others). The Vendi Score is based on Shannon entropy, which is used to measure system uncertainty. In scenarios with uneven eigenvalue distributions, the system has more information in certain feature directions, while the information in other directions is relatively small. Consequently, the overall uncertainty of the system is reduced, leading to a significant decrease in Shannon entropy. As a result, the Vendi Score underestimates the diversity of datasets with distinctive samples.
> > >
> > > >Q9: How many samples do we need for DCScore to converge?
> > >
> > > R9: Thank you for your question and for providing additional insights. The convergence setting differs from our targeted scenarios. **Our proposed method aims to evaluate already generated synthetic datasets**, such as text datasets from LLMs, which is distinct from generative model evaluations (Because our target scenario lacks ground truth values). To clarify our approach, we conducted a sample convergence experiment comparing it to Vendi Score. **We used diversity evaluation methods on datasets generated by WGAN-GP**, following the WGAN-GP [1] settings on 8, 25 Gaussian toy datasets. We present the experimental results in **Figures 7, 8 at our anonymous URL** (https://anonymous.4open.science/r/ICMLRebuttal_DCScore). Specifically, we observe that DCScore **requires 300 samples to converge** and achieves a diversity score that is closer to the actual mode number.
> > >
> > > Reference:
> > >
> > > [1] Improved Training of Wasserstein GANs, NIPS 2017.
> > >
> > > >Q10: (1) Could the authors clarify what settings they used (e.g., how many RFFs they used)? (2) I appreciate the point about the summerization part complexity, but isn't the bottleneck for the Vendi score?
> > >
> > > R10: Thank you for your questions.
> > > (1) **We set RFFs (rff_dim) to 768**, which matches the embedding dimension of BERT (the model used for extracting embeddings is bert-base-uncased). Additionally, we set the sigma parameter and batch size to 20 and 512, respectively, with the batch size being consistent with that of DCScore.
> > > (2) As shown in Table 2 of our paper, the Vendi Score has **a high complexity** ($\mathcal{O}(n^3)$) when using general kernels. This is due to the computational requirements for calculating eigenvalues. Therefore, we believe this is one of the key points where the Vendi score can be improved.
> > >
> > > >Q11: Could the authors elaborate on why this (non-shift-invariant kernels) fails to meet diversity evaluation requirements?
> > >
> > > R11:  Thank you for your question. We agree with the necessity of shift-invariant kernels in diversity evaluation. However, we would clarify our claim for three following reasons:
> > >
> > > - A general diversity evaluation method should ideally accommodate various scenarios, including those involving non-shift-invariant kernels, to ensure broader applicability.
> > > - Scenarios that may require non-shift-invariant kernels include the evaluation of diversity in image modalities, which often necessitates capturing higher-level features such as textures and edge information. In such cases, more complex kernels (e.g., polynomial kernels) are often needed to capture these nonlinear features.
> > > - **Certain non-shift-invariant kernels exhibit robustness to noise and outliers**[1], ensuring that the diversity evaluation results are not unduly affected by a small amount of anomalous data. This robustness is also essential for effective diversity evaluation.
> > >
> > > Reference:
> > >
> > > [1] Kernel approximation using analogue in-memory computing, Nature 2024.
> > >
> > > ---
> > > We once again thank you for your highly constructive comments, which have been extremely helpful in improving the quality of our work.

---

### Decision · Program_Chairs · 2025-05-01

**Decision:**

Accept (poster)

**Comment:**

The paper introduces a method for measuring the diversity in synthetic datasets in the context of classification.
There method addresses a timely question about measuring diversity in synthetic datasets with a computationally efficient procedure. The proofs of the theoretical claims appear to be sound.
The comparison with the Vendi score, however, appears to be incomplete, as it is unclear whether the proposed score is genuinely a more useful metric.